# Revisiting Decomposable Submodular Function Minimization with Incidence Relations

**Pan Li**
UIUC
panli2@illinois.edu

**Olgica Milenkovic**
UIUC
milenkov@illinois.edu

## Abstract

We introduce a new approach to decomposable submodular function minimization (DSFM) that exploits incidence relations. Incidence relations describe which variables effectively influence the component functions, and when properly utilized, they allow for improving the convergence rates of DSFM solvers. Our main results include the precise parametrization of the DSFM problem based on incidence relations, the development of new scalable alternative projections and parallel coordinate descent methods and an accompanying rigorous analysis of their convergence rates.

## 1 Introduction

A set function $F : 2^{[N]} \to \mathbb{R}$ over a ground set $[N]$ is termed submodular if for all pairs of sets $S_1, S_2 \subseteq [N]$, one has $F(S_1) + F(S_2) \geq F(S_1 \cap S_2) + F(S_1 \cup S_2)$. Submodular functions capture the ubiquitous phenomenon of diminishing marginal costs [1] and they frequently arise as part of the objective function of various machine learning optimization problems [2, 3, 4, 5, 6, 7].

Among the various submodular function optimization problems, submodular function minimization (SFM), which may be stated as $\min_{S \subseteq [N]} F(S)$, is one of the most important and commonly studied questions. The current fastest known SFM algorithm has complexity $O(N^4 \log^{O(1)} N + \tau N^3)$, where $\tau$ denotes the time needed to evaluate the submodular function [8]. Although SFM solvers operate in time polynomial in $N$, the high-degree of the underlying polynomial prohibits their use in practical large-scale settings. For this reason, a recent line of work has focused on developing scalable and parallelizable algorithms for solving the SFM problem by leveraging the property of *decomposability* [9]. Decomposability asserts that the submodular function may be written as a sum of "simpler" submodular functions that may be optimized sequentially or in parallel. Formally, the underlying problem, referred to as decomposable SFM (DSFM), may be stated as:

$$\text{DSFM:} \quad \min_{S} \sum_{r \in [R]} F_r(S), \tag{1}$$

where $F_r : 2^{[N]} \to \mathbb{R}$ is a submodular function for all $r \in [R]$. Algorithmic solutions for the DSFM problem fall into two categories, combinatorial optimization approaches [10, 11] and continuous function optimization methods [12]. In the latter setting, a crucial concept is the Lovász extension of the submodular function which is convex [13] and lends itself to a norm-regularized convex optimization framework. Prior work in continuous DSFM has focused on devising efficient algorithms for solving the convex problem and deriving matching convergence results. The best known approaches include the alternating projection (AP) methods [14, 15] and the coordinate descent (CD) methods [16].

Despite some simplifications offered through decomposibility, DSFM algorithms still suffer from scalability issues and have convergence guarantees that are suboptimal. To address the first issue, one needs to identify additional problem constraints that allow for parallel implementations. To resolve the second issue and more precisely characterize and improve the convergence rates, one needs to better understand how the individual submodular components jointly govern the global optimal solution. In both cases, it is crucial to utilize *incidence relations* that describe which subsets of variables

directly affect the value of any given component function. Often, incidences involve relatively small subsets of elements, which leads to desirable sparsity constraints. This is especially the case for min-cut problems on graphs and hypergraphs (where each submodular component involves two or several vertices) [17, 18] and MAP inference with higher-order potentials (where each submodular component involves variables corresponding to adjacent pixels) [9]. Although incidence relations have been used to parametrize the algorithmic complexity of combinatorial optimization methods for solving DSFM problems [10], they have been largely overlooked in continuous optimization methods. Some prior work considered merging decomposable parts with nonoverlapping support into one submodular function, thereby creating a coarser decomposition that may be processed more efficiently [14, 15, 16], but the accompanying algorithms were neither designed in a form that can optimally use this information nor analyzed precisely with respect to their convergence rates and merging strategies. In an independent work, Djolonga and Krause found that the variational inference problem in L-FIELD could be reduced to a DSFM problem with sparse incidence relations [19], while their analysis only worked for regular cases.

Here, we revisit two benchmark algorithms for continuous DSFM – AP and CD – and describe how to modify them to exploit incidence relations that allow for significantly improved computational complexity. Furthermore, we provide a complete theoretical analysis of the algorithms parametrized by incidence relations with respect to their convergence rates. AP-based methods that leverage incidence relations achieve better convergence rates than classical AP algorithms both in the sequential and parallel optimization scenario. The random CD method (RCDM) and accelerated CD method (ACDM) that incorporate incidence information can be parallelized. The complexity of sequential CD methods cannot be improved using incidence relations, but the convergence rate of parallel CD methods strongly depends on how the incidence relations are used for coordinate sampling: while a new specialized combinatorial sampling based on equitable coloring [20] is optimal, uniformly at random sampling produces a 2-approximation. It also leads to a greedy method that empirically outperforms random sampling. A summary of these and other findings is presented in Table 1.

|  | Prior work | | This work | |
|---|---|---|---|---|
|  | Sequential | Parallel | Sequential | Parallel |
| AP | $O(N^2 R^2)$ | $O(N^2 R^2/K)$ | $O(N\|\mu\|_1 R)$ | $O(N\|\mu\|_1 R/K)$ |
| RCDM | $O(N^2 R)$ | - | $O(N^2 R)$ | $O\left(\left(\frac{R-K}{R-1}N^2 + \frac{K-1}{R-1}N\|\mu\|_1\right) R/K\right)$ |
| ACDM | $O(NR)$ | - | $O(NR)$ | $O\left(\left(\frac{R-K}{R-1}N^2 + \frac{K-1}{R-1}N\|\mu\|_1\right)^{1/2} R/K\right)$ |

Table 1: Overview of known and new results: each entry contains the required number of iterations to achieve an $\epsilon$-optimal solution (the dependence on $\epsilon$ is the same for all algorithms and hence omitted). Here, $\|\mu\|_1 = \sum_{i\in[N]} \mu_i$, where for all $i \in [N]$, $\mu_i$ equals the number of submodular functions that involve element $i$; $K$ is a parallelization parameter that equals the number of min-norm points problems that have to be solved within each iteration.

## 2 Background, Notation and Problem Formulation

We start our exposition by reviewing several recent lines of work for solving the DSFM problem, and focus on approaches that transform the DSFM problem into a continuous optimization problem. Such approaches exploit the fact that the Lovász extension of a submodular function is convex. Without loss of generality, we tacitly assume that all submodular functions $F_r$ are normalized, i.e., that $F_r(\emptyset) = 0$ for all $r \in [R]$. Also, we define given a vector $z \in \mathbb{R}^N$ and $S \subseteq [N]$, $z(S) = \sum_{i\in S} z_i$. Then, the *base polytope* of the $r$-th submodular function $F_r$ is defined as

$$\mathcal{B}_r \triangleq \{y_r \in \mathbb{R}^N | y_r(S) \le F_r(S), \text{ for any } S \subset [N], \text{and } y_r([N]) = F_r([N])\}.$$

The *Lovász extention* [13] $f_r(\cdot) : \mathbb{R}^N \to \mathbb{R}$ of a submodular function $F_r$ is defined as $f_r(x) = \max_{y_r\in\mathcal{B}_r}\langle y_r, x\rangle$, where $\langle\cdot,\cdot\rangle$ denotes the inner product of two vectors. The DSFM problem can be solved through continuous optimization, $\min_{x\in[0,1]^N}\sum_r f_r(x)$. To counter the nonsmoothness of the objective function, a proximal formulation of a generalization of the above optimization problem is considered instead [14],

$$\min_{x\in\mathbb{R}^N} \sum_{r\in[R]} f_r(x) + \frac{1}{2}\|x\|_2^2. \tag{2}$$

As the problem (2) is strongly convex, it has a unique optimal solution, denoted by $x^*$. The exact discrete solution to the DSFM problem equals $S^* = \{i \in [N] \,|\, x_i^* > 0\}$.

For convenience, we denote the product of base polytopes as $\mathcal{B} = \otimes_{r=1}^{R} \mathcal{B}_r$, and write $y = (y_1, y_2, ..., y_R) \in \mathcal{B}$. Also, we let $A$ be a simple linear mapping $\otimes_{r=1}^{R} \mathbb{R}^N \to \mathbb{R}^N$, which given a point $a = (a_1, a_2, ..., a_R) \in \otimes_{r=1}^{R} \mathbb{R}^N$ outputs $Aa = \sum_{r \in [R]} a_r$. The AP and CD algorithms for solving (2) use the dual form of the problem, described in the next lemma.

**Lemma 2.1** ([14]). The dual problem of (2) reads as

$$\min_{a,y} \|a - y\|_2^2 \quad \text{s.t.} \quad Aa = 0, \ y \in \mathcal{B}. \tag{3}$$

Moreover, problem (3) may be written in the more compact form

$$\min_{y} \|Ay\|_2^2 \quad \text{s.t.} \quad y \in \mathcal{B}. \tag{4}$$

For both problems, the primal and dual variables are related according to $x = -Ay$. In what follows, for notational simplicity, we write $g(y) = \frac{1}{2}\|Ay\|_2^2$.

The AP [15] and RCD algorithms [16] described below provide solutions to the problems (3) and (4), respectively. They both rely on repeated projections $\Pi_{\mathcal{B}_r}(\cdot)$ onto the base polytopes $\mathcal{B}_r$, $r \in [R]$. These projections are typically less computationally intense than projections onto the complete base polytope of $F$ as they involve fewer data dimensions. The projection operation $\Pi_{\mathcal{B}_r}(\cdot)$ requires one to solve a min-norm problem by either exploiting the special forms of $F_r$ or by using the general purpose algorithm of Wolfe [21]. The complexity of the method is typically characterized by the number of required projections $\Pi_{\mathcal{B}_r}(\cdot)$.

**The AP algorithm.** Starting with $y = y^{(0)}$, iteratively compute a sequence $(a^{(k)}, y^{(k)})_{k=1,2,...}$ such that for all $r \in [R]$, $a_r^{(k)} = y_r^{(k-1)} - Ay^{(k-1)}/R$, $y_r^{(k)} = \Pi_{\mathcal{B}_r}(a_r^{(k)})$, until a stopping criteria is met.

**The RCDM algorithm.** In each iteration $k$, chose uniformly at random a subset of elements in $y$ associated with one atomic function in the decomposition (1), say the one with index $r_k$. Then, compute the sequence $(y^{(k)})_{k=1,2,...}$ according to $y_{r_k}^{(k)} = \Pi_{B_{r_k}}\left(-\sum_{r \neq r_k} y_r^{(k-1)}\right)$, $y_r^{(k)} = y_r^{(k-1)}$, for $r \neq r_k$.

Finding an $\epsilon$-optimal solution for both the AP and RCD methods requires $O(N^2 R \log(\frac{1}{\epsilon}))$ iterations. In each iteration, the AP algorithm computes the projections onto all $R$ base polytopes, while the RCDM only computes one projection. Therefore, as may be seen from Table 1, the sequential AP solver, which computes one projection in each iteration, requires $O(N^2 R^2 \log(\frac{1}{\epsilon}))$ iterations. However, the projections within one iteration of the AP method can be generated in parallel, while the projections performed in the RCDM have to be generated sequentially.

## 2.1 Incidence Relations and Related Notations

We next formally introduce one of the key concepts used in this work: *incidence relations* between elements of the ground set and the component submodular functions.

We say that an element $i \in [N]$ is *incident* to a submodular function $F$ iff there exists a $S \subseteq [N]/\{i\}$ such that $F(S \cup \{i\}) \neq F(S)$; similarly, we say that the submodular function $F$ is *incident* to an element $i$ iff $i$ is incident to $F$. To verify whether an element $i$ is incident to a submodular function $F$, one needs to verify that $F(\{i\}) = 0$ and that $F([N]) = F([N]/\{i\})$ since for any $S \subseteq [N]/\{i\}$

$$F(\{i\}) \geq F(S \cup \{i\}) - F(S) \geq F([N]) - F([N]/\{i\}).$$

Furthermore, note that if $i \in [N]$ is not incident to $F_r$, then for any $y_r \in \mathcal{B}_r$, one has $y_{r,i} = 0$. Let $S_r$ be the set of all elements incident to $F_r$. For each element $i$, denote the number of submodular functions that are incident to $i$ by $\mu_i = |\{r \in [R] : i \in S_r\}|$. We also refer to $\mu_i$ as the degree of element $i$. We find it useful to partition the set of submodular functions into different groups. Given a group $C \subseteq [R]$ of submodular functions, we define the degree of the element $i$ within $C$, $\mu_i^C$, as $\mu_i^C = |\{r \in C : i \in S_r\}|$.

We also define a skewed norm involving two vectors $w \in \mathbb{R}_{>0}^N$ and $z \in \mathbb{R}^N$ according to $\|z\|_{2,w} \triangleq \sqrt{\sum_{i \in [N]} w_i z_i^2}$. With a slight abuse of notation, for two vectors $\theta = (\theta_1, \theta_2, ..., \theta_R) \in \otimes_{r=1}^{R} \mathbb{R}_{>0}^N$

and $y \in \otimes_{r=1}^{R} \mathbb{R}^N$, we also define the norm $\|y\|_{2,\theta} \triangleq \sqrt{\sum_{r \in [R]} \|y_r\|_{2,\theta_r}^2}$. Which of the norms we refer to should be clear from the context. In addition, we let $\|\theta\|_{1,\infty} = \sum_{i \in [N]} \max_{r \in [R]:i \in S_r} \theta_{r,i}$. For a closed set $\mathcal{K} \subseteq \otimes_{r=1}^{R} \mathbb{R}^N$ and a positive vector $\theta \in \otimes_{r=1}^{R} \mathbb{R}_{>0}^N$, the distance between $y$ and $\mathcal{K}$ is defined as $d_\theta(y, \mathcal{K}) = \min\{\|y - z\|_{2,\theta} | z \in \mathcal{K}\}$. Also, given a set $\Omega \subseteq \mathbb{R}^N$, we let $\Pi_{\Omega,w}(\cdot)$ denote the projection operation onto $\Omega$ with respect to the norm $\|\cdot\|_{2,w}$.

Given a vector $w \in \mathbb{R}_{>0}^N$, we also make use of an induced vector $I(w) \in \otimes_{r=1}^{R} \mathbb{R}^N$ whose $r$-th entry satisfies $(I(w))_r = w$. It is easy to check that $\|I(w)\|_{1,\infty} = \|w\|_1$. Of special interest are induced vectors based on pairs of $N$-dimensional vectors, $\mu = (\mu_1, \mu_2, ..., \mu_N)$, $\mu^C = (\mu_1^C, \mu_2^C, ..., \mu_N^C)$. Finally, for $w, w' \in \mathbb{R}^N$, we denote the element-wise power of $w$ by $w^\alpha = (w_1^\alpha, w_2^\alpha, ..., w_N^\alpha)$, for some $\alpha \in \mathbb{R}$, and the element-wise product of $w$ and $w'$ by $w \odot w' = (w_1 w_1', w_2 w_2', ..., w_N w_N')$.

Next, recall that $x^*$ is the unique optimal solution of the problem (2) and let $\mathcal{Z} = \{\xi \in \otimes_{r=1}^{R} \mathbb{R}^N | A\xi = -x^*, \xi_{r,i} = 0, \forall i \in S_r, \forall r \in [R]\}$. Then, due to the duality relationship of Lemma 2.1, $\Xi = \mathcal{Z} \cap \mathcal{B}$ is the set of optimal solutions $\{y\}$.

# 3 Continuous DSFM Algorithms with Incidence Relations

In what follows, we revisit the AP and CD algorithms and describe how to improve their performance and analytically establish their convergence rates. Our first result introduces a modification of the AP algorithm (3) that exploits incidence relations so as to decrease the required number of iterations from $O(N^2 R)$ to $O(N\|\mu\|_1)$. Our second result is an example that shows that the convergence rates of CD algorithms [11] cannot be directly improved by exploiting the functions' incidence relations even when the incidence matrix is extremely sparse. Our third result is a new algorithm that relies of coordinate descent steps but can be parallelized. In this setting, incidence relations are essential to the parallelization process.

To analyze solvers for the continuous optimization problem (2) that exploit the incidence structure of the functions, we make use of the skewed norm $\|\cdot\|_{2,w}$ with respect to some positve vector $w$ that accounts for the fact that incidences are, in general, nonuniformly distributed. In this context, the projection $\Pi_{\mathcal{B}_r,w}(\cdot)$ reduces to solving a classical min-norm problem after a simple transformation of the underlying space which does not incur significant complexity overheads. To see this, note that in order to solve a generic min-norm point problem, one typically uses either Wolfe's algorithm (continuous) or a divide-and-conquer procedure (combinatorial). The complexity of the former is at most quadratic in $F_{r,\max} \triangleq \max_{v,S} |F_r(S \cup \{v\}) - F_r(S)|$ [22], while the complexity of the latter merely depends on $\log F_{r,\max}$ [14] (see Section A in the Supplement). It is unclear if including the weight vector $w$ into the projection procedure increases or decreases $F_{r,\max}$. In either case, given that in our derivations all elements of $w$ are contained in $[1, \max_{i \in [N]} \mu_i]$ instead of $N$ or $R$, we do not expect to see significant changes in the complexity of the projection operation. Hence, throughout the remainder of our exposition, we regard the projection operation as an oracle and measure the complexity of all algorithms in terms of the number of projections performed.

Also, observe that one may avoid computing projections in skewed-norm spaces by introducing in (2) a weighted rather than an unweighted proximal term. This gives another continuous objective that still provides a solution to the discrete problem (1). Even in this case, we can prove that the numbers of iterations used in the different methods listed Table 1 remain the same. Furthermore, by combining projections in skewed-norm spaces and weighted proximal terms, it is possible to actually reduce the number of iterations given in Table 1. However, for simplicity, we focus on the objective (2) and projections in skewed-norm spaces. Methods using weighted proximal terms with and without skewed-norm projections are analyzed in a similar manner in Section L of the Supplement.

We make frequent use of the following result which generalizes Lemma 4.1 of [11].

**Lemma 3.1.** Let $\theta \in \otimes_{r=1}^{R} \mathbb{R}_{>0}^N, w \in \mathbb{R}_{>0}^N$ be two positive vectors. Let $y \in \mathcal{B}$ and let $z$ be in the base polytope of the submodular function $F$. Then, there exists a point $\xi \in \mathcal{B}$ such that $A\xi = z$ and $\|\xi - y\|_{2,\theta} \leq \sqrt{\frac{\|\theta\|_{1,\infty}}{2}} \|Ay - z\|_1$. Moreover, $\|\xi - y\|_{2,\theta} \leq \sqrt{\frac{\|\theta\|_{1,\infty} \|w^{-1}\|_1}{2}} \|Ay - z\|_{2,w}$.

## 3.1 The Incidence Relation AP (IAP)

The following result establishes the basis of our improved AP method leveraging incidence structures.

**Lemma 3.2.** The following problem is equivalent to problem (3):

$$\min_{a,y} \|a - y\|_{2,I(\mu)}^2 \quad \text{s.t.} \quad y \in \mathcal{B}, \, Aa = 0, \text{ and } a_{r,i} = 0, \, \forall (r,i) : i \notin S_r, r \in [R]. \tag{5}$$

Let $\mathcal{A} = \{a \in \otimes_{r=1}^{R} \mathbb{R}^N | Aa = 0, a_{r,i} = 0, \, \forall (r,i) : i \notin S_r\}$ and $\mathcal{A}' = \{a \in \otimes_{r=1}^{R} \mathbb{R}^N | Aa = 0\}$. The AP algorithm for problem (5) consists of alternatively computing projections between $\mathcal{A}$ and $\mathcal{B}$, as opposed to those between $\mathcal{A}'$ and $\mathcal{B}$ used in the problem (3). However, as already pointed out, unlike for the classical AP problem (3), the distance in (5) is not Euclidean, and hence the projections may not be orthogonal.

The IAP method for solving (5) proceeds as follows. We begin with $a = a^{(0)} \in \mathcal{A}$, and iteratively compute a sequence $(a^{(k)}, y^{(k)})_{k=1,2,\dots}$ as follows: for all $r \in [R]$, $y_r^{(k)} = \Pi_{\mathcal{B}_r,\mu}(a_r^{(k)})$, $a_{r,i}^{(k)} = y_{r,i}^{(k-1)} - \mu_i^{-1}(Ay^{(k-1)})_i$, $\forall \, i \in S_r$. The key difference between the AP and IAP algorithms is that the latter effectively removes "irrelevant" components of $y_r$ by fixing the irrelevant components of $a$ to 0. In the AP method of Nishihara [15], these components are never zero as they may be "corrupted" by other components during AP iterations. Removing irrelevant components results in projecting $y$ into a subspace of lower dimensions, which significantly accelerates the convergence of IAP.

Figure 1: Illustration of the IAP method for solving problem (5): The space $\mathcal{A}$ is a subspace of $\mathcal{A}'$, which leads to faster convergence of the IAP method when compared to AP.

The analysis of the convergence rate of the IAP method follows a similar outline as that used to analyze (3) in [15]. Following Nishihara et al. [15], we define the following parameter that plays a key role in determining the rate of convergence of the AP algorithm, $\kappa_* \triangleq \sup_{y \in \mathcal{Z} \cup \mathcal{B}/\Xi} \frac{d_{I(\mu)}(y,\Xi)}{\max\{d_{I(\mu)}(y,\mathcal{Z}), d_{I(\mu)}(y,\mathcal{B})\}}$.

**Lemma 3.3** ([15]). If $\kappa_* < \infty$, the AP algorithm converges linearly with rate $1 - \frac{1}{\kappa_*^2}$. At the $k$-th iteration, the algorithm outputs a value $y^{(k)}$ that satisfies

$$d_{I(\mu)}(y^{(k)}, \Xi) \leq 2 d_{I(\mu)}(y^{(0)}, \Xi) \left(1 - \frac{1}{\kappa_*^2}\right)^k.$$

To apply the above lemma in the IAP setting, one first needs to establish an upper bound on $\kappa_*$. This bound is given in Lemma 3.4 below.

**Lemma 3.4.** The parameter $\kappa_*$ is upper bounded as $\kappa_* \leq \sqrt{N\|\mu\|_1/2} + 1$.

By using the above lemma and the bound on $\kappa_*$, one can establish the following convergence rate for the IAP method.

**Theorem 3.5.** After $O(N\|\mu\|_1 \log(1/\epsilon))$ iterations, the IAP algorithm for solving problem (5) outputs a pair of points $(a, y)$ that satisfies $d_{I(\mu)}(y, \Xi) \leq \epsilon$.

Note that in practice, one often has $\|\mu\|_1 \ll NR$, which shows that the convergence rate of the AP method for solving the DSBM problem may be significantly improved.

## 3.2 Sequential Coordinate Descent Algorithms

Unlike the AP algorithm, the CD algorithms by Ene et al. [16] remain unchanged given (4). Our first goal is to establish whether the convergence rate of the CD algorithms can be improved using a parameterization that exploits incidence relations.

The convergence rate of CD algorithms is linear if the objective function is component-wise smooth and $\ell$-strong convex. In our case, $g(y)$ is component-wise smooth as for any $y, z \in \mathcal{B}$ that only differ

in the $r$-th block (i.e., $y_r \neq z_r$, $y_{r'} = z_{r'}$ for $r' \neq r$), one has

$$\|\nabla_r g(y) - \nabla_r g(z)\|_2 \leq \|y - z\|_2. \tag{6}$$

Here, $\nabla_r g$ denotes the gradient vector associated with the $r$-th block.

**Definition 3.6.** We say that the function $g(y)$ is $\ell$-*strongly convex* in $\|\cdot\|_{2,}$, if for any $y \in \mathcal{B}$

$$g(y^*) \geq g(y) + \langle \nabla g(y), y^* - y \rangle + \frac{\ell}{2}\|y^* - y\|_2^2, \text{ or equivalently,} \quad \|Ay - Ay^*\|_2^2 \geq \ell\|y^* - y\|_2^2,$$

where $y^* = \arg\min_{z \in \Xi} \|z - y\|_2^2$. Moreover, we let $\ell_* = \sup\{\ell : g(y) \text{ is } \ell\text{-strongly convex in } \|\cdot\|_2\}$.

Note that the above definition essentially establishes a form of weak-strong convexity [23]. Then, using standard analytical tools for CD algorithms [24], we can prove the following result [16].

**Theorem 3.7.** The RCDM for problem (4) outputs a point $y$ that satisfies $\mathbb{E}[g(y)] \leq g(y^*) + \epsilon$ after $O(\frac{R}{\ell_*}\log(1/\epsilon))$ iterations. The ACDM applied to the problem (4) outputs a point $y$ that satisfies $\mathbb{E}[g(y)] \leq g(y^*) + \epsilon$ after $O(\frac{R}{\sqrt{\ell_*}}\log(1/\epsilon))$ iterations.

To precisely characterize the convergence rate, we need to find an accurate estimate of $\ell_*$. Ene et al. [11] derived $\ell_* \geq \frac{1}{N^2}$ without taking into account the incidence structure. As sparse incidence side information improves the performance of the AP method, it is of interest to determine if the same can be accomplished for the CD algorithms. Example 3.1 establishes that this is not possible in general if one only relies on $\ell_*$.

*Example* 3.1. Consider a DSFM problem with a extremely sparse incidence structure with $|S_r| = 2$. More precisely, let $N = 2n+1$, $R = 2n$, and $\|\mu\|_1 = \sum_{r \in [R]} |S_r| = 4n \ll NR$. Let $F_r$ be incident to the elements $\{r, r+1\}$, for all $r \in [R]$, and be such that $F_r(\{r\}) = F_r(\{r+1\}) = 1$, $F_r(\emptyset) = F_r(\{r, r+1\}) = 0$. Then, $\ell_* < \frac{7}{N^2}$.

Note that the optimal solution of problem (4) for this particular setting equals $y^* = 0$. Let us consider a point $y \in \mathcal{B}$ specified as follows. First, due to the given incidence relations, the block $y_r$ has two components corresponding to the elements indexed by $r$ and $r+1$. For any $r \in [R]$,

$$y_{r,r} = -y_{r,r+1} = \begin{cases} \frac{r}{n} & r \leq n, \\ \frac{2n+1-r}{n} & r \geq n+1. \end{cases} \tag{7}$$

Therefore, $g(y) = \frac{1}{n}$, $\|y\|_2^2 > \frac{4}{3}n$, which results in $\ell_* < \frac{3}{2n^2} \leq \frac{7}{N^2}$ for all $n \geq 3$.

Example 3.1 only illustrates that an important parameter of CDMs cannot be improved using incidence information; but this does not necessarily imply that a sequential RCDM that uses incidence structures cannot offer better convergence rates than $O(N^2 R)$. In Section E of the Supplement, we present additional experimental evidence that supports our observation, using the setting of Example 3.1.

As a final remark, note that Nishihara et al. [15] also proposed a lower bound that does not make use of sparse incidence structures and only works for the AP method.

## 3.3 New Parallel CD methods

In what follows, we propose two CDMs which rely on parallel projections and incidence relations.

The following observation is key to understanding the proposed approach. Suppose that we have a nonempty group of blocks $C \subseteq [R]$. Let $y, h \in \otimes_{r=1}^R \mathbb{R}^N$. If $h_{r,i}$ is nonzero only for block $r \in C$ and $i \in S_r$, then,

$$g(y + h) = g(y) + \langle \nabla g(y), h \rangle + \frac{1}{2}\|Ah\|_2^2 \leq g(y) + \sum_{r \in C} \langle \nabla_r g(y), h_r \rangle + \sum_{r \in C} \frac{1}{2}\|h_r\|_{2,\mu^C}^2. \tag{8}$$

Hence, for all $r \in C$, if we perform projections onto $\mathcal{B}_r$ with respect to the norm $\|\cdot\|_{2,\mu^C}$ simultaneously in each iteration of the CDM, convergence is guaranteed as the value of the objective function remains bounded. The smaller the components of $\mu^C$, the faster the convergence. Note that the components of $\mu^C$ are the numbers of incidence relations of elements restricted to the set $C$. Hence, in each iteration, blocks that ought to be updated in parallel are those that correspond to submodular functions that have supports with smallest possible intersections.

One can select blocks that are to be updated in parallel in a combinatorially specified fashion or in a randomized fashion, as dictated by what we call an $\alpha$-proper distribution. To describe our parallel RCDM, we first introduce the notion of an $\alpha$-proper distribution.

**Definition 3.8.** Let $P$ be a distribution used to sample a group of $C$ blocks. Define $\theta^P = (\theta_1^P, \theta_2^P, ..., \theta_R^P)$ such that for $r \in [R]$, $\theta_r^P \triangleq \mathbb{E}_{C \sim P}\left[\mu^C | r \in C\right]$. We say that $P$ is an $\alpha$-proper distribution, if for any $r \in [R]$ and a given $\alpha \in (0, 1)$, we have $\mathbb{P}(r \in C) = \alpha$.

We are now ready to describe the parallel RCDM algorithm – Algorithm 1; the description of the parallel ACDM is postponed to Section J of the Supplement.

---

**Algorithm 1: Parallel RCDM for Solving** (4)

**Input**: $\mathcal{B}, \alpha$

0: Initialize $y^{(0)} \in \mathcal{B}$, $k \leftarrow 0$

1: Do the following steps iteratively until the dual gap $< \epsilon$:

2:     Sample $C_{i_k}$ using some $\alpha$-proper distribution $P$

3:     For $r \in C_{i_k}$:

4:         $y_r^{(k+1)} \leftarrow \Pi_{\mathcal{B}_r, \theta_r^P}(y_r^{(k)} - (\theta_r^P)^{-1} \odot \nabla_r g(y^{(k)}))$

5:     Set $y_r^{(k+1)} \leftarrow y_r^{(k)}$ for $r \notin C_{i_k}$, $k \leftarrow k + 1$

6: Output $y^{(k)}$

---

Next, we establish strong convexity results for the space $\| \cdot \|_{2, \theta^P}$ by invoking Lemma 3.1.

**Lemma 3.9.** For any $y \in \mathcal{B}$, let $y^* = \arg\min_{\xi \in \Xi} \|\xi - y\|_{2, \theta^P}^2$. Then,

$$\|Ay - Ay^*\|_2^2 \geq \frac{2}{N\|\theta^P\|_{1, \infty}} \|y - y^*\|_{2, \theta^P}^2.$$

The convergence rate of Algorithm 1 is established in the next theorem.

**Theorem 3.10.** At each iteration of Algorithm 1, $y^{(k)}$ satisfies

$$\mathbb{E}\left[g(y^{(k)}) - g(y^*) + \frac{1}{2}d_{\theta^P}^2(y^k, \xi)\right] \leq \left[1 - \frac{4\alpha}{(N\|\theta^P\|_{1, \infty} + 2)}\right]^k \left[g(y^{(0)}) - g(y^*) + \frac{1}{2}d_{\theta^P}^2(y^0, \xi)\right].$$

The parameter $N\|\theta^P\|_{1, \infty}$ is obtained by combining the strong convexity constant and the properties of the sampling distribution $P$. Small values of $\|\theta^P\|_{1, \infty}$ ensure better convergence rates, and we next bound this value.

**Lemma 3.11.** For any $\alpha$-proper distribution $P$ and an element $i \in [N]$, $\max_{r \in [R]: i \in S_r} \theta_{r, i}^P \geq \max\{\alpha\mu_i, 1\}$. Consequently, $\|\theta^P\|_{1, \infty} \geq \max\{\alpha\|\mu\|_1, N\}$.

Without considering incidence relations, i.e., by setting $\|\mu\|_1 = NR$, one always has $\|\theta^P\|_{1, \infty} \geq \alpha NR$, which shows that parallelization cannot improve the convergence rate of the RCDM.

The next lemma characterizes an achievable $\|\theta^P\|_{1, \infty}$ obtained by choosing $P$ to be a uniform distribution, which, when combined with Theorem 3.10, proves the result of the last column in Table 1.

**Lemma 3.12.** If $C$ is a set of size $0 < K \leq R$ obtained by sampling the $K$-subsets of $[R]$ uniformly at random, then $\theta_r^P = \frac{K-1}{R-1}\mu + \frac{R-K}{R-1}\mathbf{1}$. Moreover, $\|\theta^P\|_{1, \infty} = \frac{K-1}{R-1}\|\mu\|_1 + \frac{R-K}{R-1}N$.

Comparing Lemma 3.11 and Lemma 3.12, we see that the $\|\theta^P\|_{1, \infty}$ achieved by sampling uniformly at random is at most a factor of two of the lower bound since $\alpha = K/R$. A natural question is if it is possible to devise a better sampling strategy. This question is addressed in Section K of the Supplement, where we related the sampling problem to equitable coloring [20]. By using Hajnal-Szemerédi's Theorem [25], we derived a sufficient condition under which an $\alpha$-proper distribution $P$ that achieves the lower bound in Lemma 3.11 can be found in polynomial time. We also described a greedy algorithm for minimizing $\|\theta^P\|_{1, \infty}$ that empirically convergences faster than sampling uniformly at random.

## 4 Experiments

In what follows, we illustrate the performance of the newly proposed DSFM algorithms on a benchmark datasets used for MAP inference in image segmentation [9] and used for semi-supervised

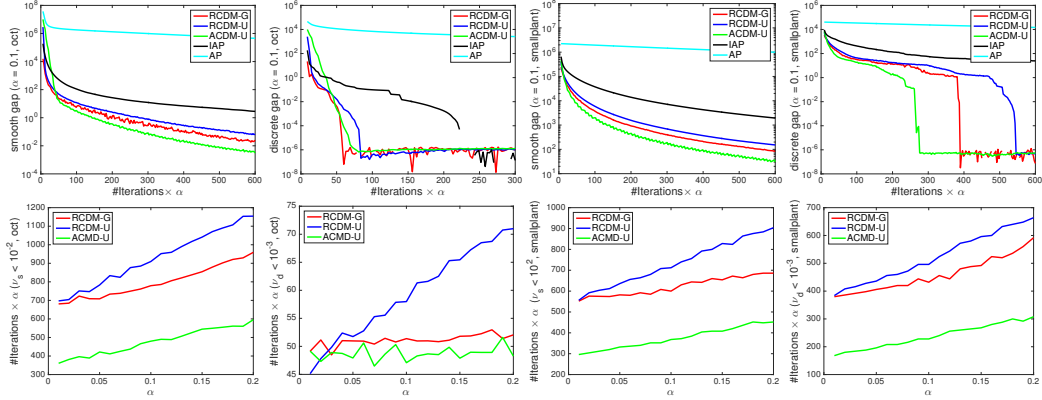

Figure 2: Image segmentation example. First row: Gap vs the number of iterations $\times\alpha$. Second row: The number of iterations $\times\alpha$ vs $\alpha$. Here, $\alpha$ is the parallelization parameter, while $K = \alpha R$ equals the number of projections that have to be computed in each iteration.

learning over graphs [1]. More experiments on semi-supervised learning over hypergraphs can be found in Section M of the Supplement.

In all the experiments, we evaluated the convergence rate of the algorithms by using the smooth duality gap $\nu_s$ and the discrete duality gap $\nu_d$. The primal problem solution equals $x = -Ay$ so that the smooth duality gap can be computed according to $\nu_s = \sum_r f_r(x) + \frac{1}{2}\|x\|^2 - (-\frac{1}{2}\|Ay\|^2)$. Moreover, as the level set $S_\lambda = \{v \in [N]|x_v > \lambda\}$ can be easily found based on $x$, the discrete duality gap can be written as $\nu_d = \min_\lambda F(S_\lambda) - \sum_{v \in [N]} \min\{-x_v, 0\}$.

**MAP inference**. We used two images – *oct* and *smallplant* – adopted from [14][2]. The images comprise $640 \times 427$ pixels so that $N = 273,280$. The decomposable submodular functions are constructed following a standard procedure. The first class of functions arises from the 4-neighbor grid graph over the pixels. Each edge corresponds to a pairwise potential between two adjacent pixels $i, j$ that follows the formula $\exp(-\|v_i - v_j\|_2^2)$, where $v_i$ is the RGB color vector of pixel $i$. We split the vertical and horizontal edges into rows and columns that result in $639 + 426 = 1065$ components in the decomposition. Note that within each row or each column, the edges have no overlapping pixels, so the projections of these submodular functions onto the base polytopes reduce to projections onto the base polytopes of edge-like submodular functions. The second class of submodular functions contain clique potentials corresponding to the superpixel regions; specifically, for region $r$, $F_r(S) = |S|(|S_r| - |S|)$ [26]. These functions give another $500$ decomposition components. We apply the divide and conquer method in [14] to compute the projections required for this type of submodular functions. Note that in each experiment, all components of the submodular function are of nearly the same size, and thus the projections performed for different components incur similar computational costs. As the projections represent the primary computational units, for comparative purposes we use the number of iterations (similarly to [14, 16]).

We compared five algorithms: RCDM with a sampling distribution $P$ found by the greedy algorithm (RCDM-G), RCDM with uniform sampling (RCDM-U), ACDM with uniform sampling (ACDM-U), AP based on (5) (IAP) and AP based on (3) (AP). Figure 2 depicts the results. In the first row, we compared the convergence rates of different algorithms for a fixed parallelization parameter $\alpha = 0.1$. The values on the horizontal axis correspond to # iterations $\times\alpha$, the total number of projections performed divided by $R$. The results are averaged over 10 independent experiments. We observe that the CD-based methods outperform AP-based methods, and that ACDM-U is the best performing CD-based method. IAP significantly outperforms AP. Similarly, RCDM-G outperforms RCDM-U. We also investigated the relationship between the number of iterations and the parameter $\alpha$. We recorded the number of iterations needed to achieve a smooth and discrete gap below a certain given threshold. The results are shown in the second row of Figure 2. We did not plot the curves for the AP-based methods as they are essentially horizontal lines. Among the CD-based methods, ACDM-U performs best. RCDM-G offers a much better convergence rate than RCDM-U since the sampling probability $P$ produced by the greedy algorithm leads to a smaller value of $\|\theta^P\|_{1,\infty}$ compared to

[1]The code for this work can be found in https://github.com/lipan00123/DSFM-with-incidence-relations.

[2]Downloaded from the website of Professor Stefanie Jegelka: http://people.csail.mit.edu/stefje/code.html

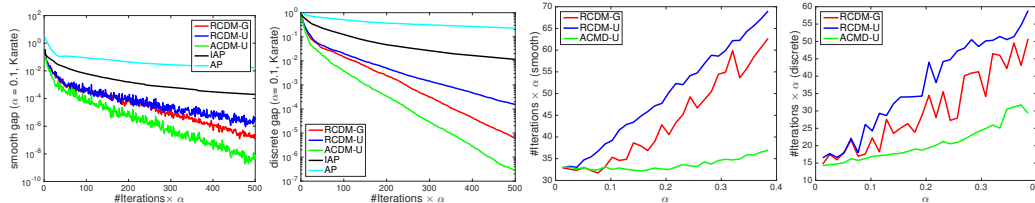

Figure 3: Zachary's Karate Club. Left two: Gap vs the number of iterations $\times\alpha$. Right two: The number of iterations $\times\alpha$ vs $\alpha$. Here, $\alpha$ is the parallelization parameter, while $K = \alpha R$ equals the number of projections that have to be computed in each iteration.

uniform sampling. The reason behind this finding is that the supports of the components in the decomposition are localized, which makes the sampling $P$ obtained from the greedy algorithm highly effective. For RCDM-U, the total number of iterations increases almost linearly with $\alpha (= K/R)$, which confirms the results of Lemma 3.12.

Note that in the above examples of MAP inference, another way to decompose the submodular functions is available: as there are three natural layers of non-overlapping incidence sets, we can merge all vertical edges, all horizontal edges, and all superpixel regions into three components respectively. Then, each of this component is incident to all pixels, and the derived results in this work will reduce to those of the former works [14, 16]. However, such a way to decompose submodular function strongly depends on the particular structure and thus is not general for DSFM problems. The following example on semi-supervised learning over graphs does not contain natural layers for decomposition.

**Semi-supervised learning**. We tested our algorithms over the dataset of Zachary's karate club [27]. This dataset is used as a benchmark example for evaluating semisupervised learning algorithms over graphs [28]. It includes $N = 34$ vertices and $R = 78$ submodular functions in the decomposition, each corresponding to one edge in the network. The objective function of both semi-supervised learning problems may be written as

$$\min_x \tau \sum_{r\in[R]} f_r(x) + \frac{1}{2}\|x - x_0\|_2^2 \tag{9}$$

where $\tau$ is a parameter that needs to be tuned, and $x_0 \in \{-1, 0, 1\}^N$, so that the nonzero components correspond to the labels that are known a priori. In our case, as we are only concerned with the convergence rate of the algorithm, we fix $\tau = 0.1$. In the experiments for Zachary's karate club, we set $x_0(1) = 1$, $x_0(34) = -1$ and let all other components of $x_0$ be equal to zero.

Figure 3 shows the results of the experiments pertaining to Zachary's karate club. In the left two subfigures, we compared the convergence rates of different algorithms for a fixed parallelization parameter $\alpha = 0.1$. The values on the horizontal axis correspond to # iterations $\times \alpha$, the total number of projections performed divided by $R$. In the right two subfigures, we controlled the numbers of projections executed within one iteration by tuning the parameter $\alpha$ and recorded the number of iterations needed to achieve smooth/discrete gaps below $10^{-3}$. The values depicted on the vertical axis correspond to # iterations $\times\alpha$, describing the total number of projections needed to achieve the given accuracy. In all cases, we see the similar tendency to that of the MAP inference. As may be seen, AP-based methods require more projections than CD-based methods, but IAP consistently outperforms AP, which is consistent with our theoretical results. Among the CD-based methods, ACDM-U offers the best performance in general, and RCDM-G slightly outperforms RCDM-U, since the greedy algorithm used for sampling produces a smaller $\|\theta^P\|_{1,\infty}$ than uniform sampling. As the AP-based methods are completely parallelizable, and increasing the parameter $\alpha$ does not increase the total number of projections. However, for RCDM-U, the total number of iterations required increases almost linearly with $\alpha$, which is supported by the result in Lemma 3.12. The performance curve for RCDM-G exhibits large oscillations due to the discrete problem component, needed for finding a balanced partition.

# 5 Acknowledgement

The authors gratefully acknowledge many useful suggestions by the reviewers. This work was supported in part by the NSF grant CCF 15-27636, the NSF Purdue 4101-38050 and the NFT STC center Science of Information.

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
