[Supplementary Material]

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

# Supplement

## A    Discrete Optimization Approach for Computing the Projections $\Pi_{\mathcal{B}_r,w}(\cdot)$

The following Lemma A.1 describes how the projections $\Pi_{\mathcal{B}_r,w}(\cdot)$ can be performed via discrete optimization. Discrete methods are especially useful when $F_r(S)$ is concave in $|S|$, as in this case they have much smaller complexity than the min-norm algorithm of Wolfe [21].

**Lemma A.1.** The optimization problem $\min_{y_r \in \mathcal{B}_r} \|z - y_r\|_{2,w}^2$ is the dual of the problem $\min_{x \in \mathbb{R}^N} f_r(x) - \langle x, z \rangle + \frac{1}{2} \|x\|_{2,w^{-1}}^2$. A solution with coordinate accuracy $\epsilon$ for the latter setting can be obtained by solving the discrete problem

$$\min_S F_r(S) - z(S) + \lambda \sum_{i \in S_r \cap S} w_i^{-1},$$

where

$$\lambda \in \left[ \min_{i \in [N]} [-F_r(\{i\}) + z(\{i\})] w_i, \ \max_{i \in [N]} [F_r([N]/\{i\}) - F_r([N]) + z(\{i\})] w_i \right],$$

at most $\min\{|S_r|, \log 1/\epsilon\}$ times. The parameter $\lambda$ is chosen based on a binary search procedure which requires solving the discrete problem $O(\log 1/\epsilon)$ times.

*Proof.* The first statement follows from $f_r(x) = \max_{y_r \in \mathcal{B}_r} \langle y_r, x \rangle$ and some simple algebra. The second claim follows from the divide and conquer algorithm described in Appendix B of [14]. □

## B    Proof of Lemma 3.1

The first part of the proof follows along the same line as the corresponding proof of Ene et al. [11] which is based on a submodular auxiliary graph and the path-augmentation algorithm [29], described in what follows.

Let $G = (V, E)$ be a directed graph such that the vertex set $V$ corresponds to the elements in $[N]$, and where the arc set may be written as $E = \cup_{r \in [R]} E_r$, with $E_r$ corresponding to a complete directed graph on the set of elements $S_r$ incident to $F_r$. With each arc $(u, v)$, we associate a capacity value based on a $y' \in \mathcal{B}$ according to $c(u,v) \triangleq \min\{f_r(S) - y_r'(S) : S \subseteq S_r, u \in S, v \notin S\}$.

Next, we consider a procedure termed path augmentations over $G$ that sequentially transforms $y'$ from $y' = y$ to a point in $\mathcal{B}$ that satisfies $Ay' = z$; the vector $y'$ is kept within $\mathcal{B}$ during the whole procedure. Let the set of source and sink nodes of the graph be defined as $N \triangleq \{v \in [N] | (Ay')_v < z_v\}$ and $P \triangleq \{v \in [N] | (Ay')_v > z_v\}$, where $z$ is as defined in the statement of the lemma. If $N = P = \emptyset$, we have $Ay' = z$. It can be shown that there always exists a directed path with positive capacity from $N$ to $P$ unless $N = P = \emptyset$ [11]. In each step, we find the shortest directed path, denoted by $\mathcal{Q}$, with positive capacity from $N$ to $P$. For each arc $(u, v)$ in $\mathcal{Q}$, if the arc belongs to $E_r$, we set $y_{r,u}' \leftarrow y_{r,u}' + \rho$, $y_{r,v}' \leftarrow y_{r,v}' - \rho$, where $\rho$ denotes the smallest capacity of any arc in $\mathcal{Q}$. This procedure ensures that $y' \in \mathcal{B}$ and that the procedure terminates in a finite number of steps, with $N = P = \emptyset$ [29].

The second part of the proof differs from the derivations of Ene et al. [11]. Suppose that $\{y'^{(0)} = y, y'^{(1)}, ..., y'^{(t)}\}$ is a sequence such that $y'^{(i)}$ equals the vector $y'$ after the $i$-th step of the above procedure. We also assume that $Ay'^{(t)} = z$, implying that the algorithm terminated at step $t$. Hence, the point $y'^{(t)}$ is the desired value of $\xi$. During path-augmentation, no element appears in more than two updated arcs. Hence,

$$\|y'^{(i)} - y'^{(i-1)}\|_{2,\theta} \leq \sqrt{2 \sum_v \max_{r \in [R]: v \in S_r} \theta_{r,v}} \rho = \sqrt{2\|\theta\|_{1,\infty}} \rho.$$

As $\|Ay'^{(i)} - Ay'^{(i-1)}\|_1 = 2\rho$, we have

$$\|y'^{(i)} - y'^{(i-1)}\|_{2,\theta} \leq \sqrt{\frac{\|\theta\|_{1,\infty}}{2}} \|Ay'^{(i)} - Ay'^{(i-1)}\|_1.$$

An important observation is that during the path-augmentation procedure, for each component $v \in [N]$, the updated sequence $\{(Ay'^{(i)})_v\}_{i=1,2,\ldots,t}$ converges monotonically to $z_v$. Hence, $\|Ay'^{(t)} - Ay'^{(0)}\|_1 = \sum_{i=1}^{t} \|Ay'^{(i)} - Ay'^{(i-1)}\|_1$. By using the triangle inequality for the norm $\| \cdot \|_{2,\theta}$, we obtain

$$\sqrt{\frac{\|\theta\|_{1,\infty}}{2}} \|z - Ay\|_1 = \sqrt{\frac{\|\theta\|_{1,\infty}}{2}} \|Ay'^{(t)} - Ay'^{(0)}\|_1 \geq \sum_{i=1}^{t} \|y'^{(i)} - y'^{(i-1)}\|_{2,\theta}$$

$$\geq \|y'^{(t)} - y'^{(0)}\|_{2,\theta} = \|y'^{(t)} - y\|_{2,\theta}.$$

Invoking the Cauchy-Schwarz inequality establishes $\|z - Ay\|_1 \leq \sqrt{\|w^{-1}\|_1} \|z - Ay\|_{2,w}$, which concludes the proof.

## C   Proof for Lemma 3.2

The equivalence between problem (5) and problem (4) is easy to establish, as $y$ is obtained from $y'$ by simply removing its zero components. The second statement is proved as follows:

$$\min_{y \in \mathcal{B}} \min_{a: Aa=0, a_{r,i}=0, \forall (r,i): i \notin S_r} \frac{1}{2} \|y - a\|_{2, I(\mu)}^2$$

$$= \min_{y \in \mathcal{B}} \min_{a: a_{r,i}=0, \forall (r,i): i \notin S_r} \max_{\lambda \in \mathbb{R}^N} \frac{1}{2} \|y - a\|_{2, I(\mu)}^2 - \langle \lambda, Aa \rangle$$

$$\overset{1)}{=} \min_{y \in \mathcal{B}} \max_{\lambda \in \mathbb{R}^N} \min_{a \in \otimes_{r=1}^{R} R^N} \frac{1}{2} \sum_{r \in [R]} \sum_{i \in S_r} [\mu_i (y_{r,i} - a_{r,i})^2 - 2\lambda_i a_{r,i}]$$

$$= \min_{y \in \mathcal{B}} \max_{\lambda \in \mathbb{R}^N} \frac{1}{2} \sum_{r \in [R]} \sum_{i \in S_r} [\mu_i^{-1} \lambda_i^2 - 2\lambda_i (\mu_i^{-1} \lambda_i + y_{r,i})]$$

$$= \min_{y \in \mathcal{B}} \max_{\lambda \in \mathbb{R}^N} \frac{1}{2} \sum_{r \in [R]} \sum_{i \in S_r} (-\mu_i^{-1} \lambda_i^2 - 2\lambda_i y_{r,i})$$

$$\overset{2)}{=} \min_{y \in \mathcal{B}} \max_{\lambda \in \mathbb{R}^N} -\frac{1}{2} \|\lambda\|_2^2 - \langle \lambda, Ay \rangle$$

$$= \min_{y \in \mathcal{B}} \|Ay\|_2^2,$$

where 1) is obtained using the incidence relations $y_{r,i} = a_{r,i} = 0$ if $i \notin S_r$ and 2) holds because $\mu_i = |\{r \in [R] | i \in S_r\}|$. The optimal $y, a, \lambda$ satisfy $a_{r,i} = y_{r,i} + \mu_i^{-1} \lambda_i$ for all $i \in S_r$, $r \in [R]$ and $\lambda = -Ay$.

## D   Proof of Lemma 3.4

First, consider a $y \in \mathcal{B}/\Xi$. We have $d_{I(\mu)}(y, \mathcal{Z}) = \|Ay + x^*\|_2$, since

$$\frac{1}{2} d_{I(\mu)}(y, \mathcal{Z})^2 = \min_{a \in \mathcal{Z}} \frac{1}{2} \|y - a\|_{2, I(\mu)}^2$$

$$= \min_{a: a_{r,i}=0, \forall (r,i): i \notin S_r} \max_{\lambda \in \mathbb{R}^N} \frac{1}{2} \|y - a\|_{2, I(\mu)}^2 - \langle \lambda, Aa + x^* \rangle$$

$$\overset{1)}{=} \max_{\lambda \in \mathbb{R}^N} \min_{a \in \otimes_{r=1}^{R} R^N} \frac{1}{2} \sum_{r \in [R]} \sum_{i \in S_r} [\mu_i (y_{r,i} - a_{r,i})^2 - 2\lambda_i a_{r,i}] - \langle \lambda, x^* \rangle$$

$$= \max_{\lambda \in \mathbb{R}^N} \frac{1}{2} \sum_{r \in [R]} \sum_{i \in S_r} [-\mu_i^{-1} \lambda_i^2 - \lambda_i y_{r,i}] - \langle \lambda, x^* \rangle$$

$$\overset{2)}{=} \max_{\lambda \in \mathbb{R}^N} -\frac{1}{2} \|\lambda\|_2^2 - \lambda^T (Ay + x^*)$$

$$= \frac{1}{2} \|Ay + x^*\|_2^2.$$

where 1) is obtained using the incidence relations $y_{r,i} = a_{r,i} = 0$ if $i \notin S_r$ and 2) holds because $\mu_i = |\{r \in [R]|i \in S_r\}|$. Based on Lemma 3.1, we know that there exists a $\xi \in \mathcal{B}$ such that $A\xi = -x^*$ and

$$\|y - \xi\|_{2,I(\mu)} \leq \sqrt{\frac{N\|I(\mu)\|_{1,\infty}}{2}}\|Ay + x^*\|_2 = \sqrt{\frac{N\|\mu\|_1}{2}}\|Ay + x^*\|_2.$$

Therefore, $\kappa(y) = \frac{d_{I(\mu)}(y,\Xi)}{d_{I(\mu)}(y,\mathcal{Z})} \leq \sqrt{\frac{N\|\mu\|_1}{2}}$.

Next, consider a $y \in \mathcal{Z}/\Xi$. As $\mathcal{B}$ is compact, there exists a $y' \in \mathcal{B}$ that achieves $d_{I(\mu)}(y,\mathcal{B}) = \|y - y'\|_{2,I(\mu)}$. Based on Lemma 3.1, we also know that there exists a $\xi \in \mathcal{B}$ such that $A\xi = -x^*$ and

$$\|\xi - y'\|_{2,I(\mu)} \leq \sqrt{\frac{\|I(\mu)\|_{1,\infty}}{2}}\|Ay' + x^*\|_1 = \sqrt{\frac{\|\mu\|_1}{2}}\|Ay' + x^*\|_1.$$

Moreover, we have

$$\|Ay' + x^*\|_1 = \|Ay' - Ay\|_1 \leq \|y' - y\|_1 = \sum_{v\in[N]}\sum_{r:v\in S_r}|y'_{r,v} - y_{r,v}|$$

$$\leq \sum_{v\in[N]}\left[\mu_v\sum_{r:v\in S_r}(y'_{r,v} - y_{r,v})^2\right]^{\frac{1}{2}} \leq \sqrt{N}\|y' - y\|_{2,I(\mu)}.$$

As $\xi \in \Xi$, it holds that $d_{I(\mu)}(y,\Xi) \leq \|\xi - y\|_{2,I(\mu)} \leq \|y' - y\|_{2,I(\mu)} + \|y' - \xi\|_{2,I(\mu)}$. In addition, as

$$\|y' - \xi\|_{2,I(\mu)} \leq \sqrt{\frac{\delta^s}{2}}\|Ay' + x^*\|_1 \leq \sqrt{\frac{N\|\mu\|_1}{2}}\|y' - y\|_{2,I(\mu)},$$

we know that $d_{I(\mu)}(y,\Xi) \leq (1 + \sqrt{\frac{N\|\mu\|_1}{2}})\|y' - y\|_{2,I(\mu)}$. Therefore,

$$\kappa(y) = \frac{d_{I(\mu)}(y,\Xi)}{d_{I(\mu)}(y,\mathcal{B})} \leq \left(1 + \sqrt{\frac{N\|\mu\|_1}{2}}\right),$$

which concludes the proof.

## E   Simulation for Example 3.1

We provide additional empirical evidence that the convergence result suggested by the bound on $\ell_* \leq \frac{7}{N^2}$ is correct. We constructed a DSFM problem following Example 3.1 and initialized $y$ according to equation (7). We used the number of iterations $k$ required to attain $g(y^{(k)}) \leq \epsilon g(y^{(0)})$ as a measure for the speed of convergence. We ran the simulations for $n \in [5, 50]$ and averaged the results for each $n$ over 10 independent runs. Figure 4 shows the results. The values next to the curves are their slopes obtained via a linear regression involving $\ln(\# \text{Iterations}) \sim \ln(N)$. As the accuracy threshold increases, the slope approaches the value 3, which indicates that the required number of iterations equals $O(N^2 R)$.

## F   Proof of Lemma 3.9

Choose $z = Ay^*$ in Lemma 3.1. Then, there is a $\xi \in \mathcal{B}$ such that $\|Ay - Ay^*\|^2 \geq \frac{2}{N\|\theta^P\|_{1,\infty}}\|y - \xi\|_{2,\theta^P}^2$. Moreover as $A\xi = z = Ay^* = -x^*$, we also have $\xi \in \Xi$. Therefore, $\|y - \xi\|_{2,\theta^P}^2 \geq \|y - y^*\|_{2,\theta^P}^2$, which concludes the proof.

## G   Proof for Theorem 3.10

First, given a group of blocks $C$ and $y \in \otimes_{r=1}^R \mathbb{R}^N$, we define $y_{[C]} \in \otimes_{r=1}^R \mathbb{R}^N$ as

$$(y_{[C]})_r = \begin{cases} y_r & \text{if } r \in C, \\ 0 & \text{if } r \notin C. \end{cases}$$

The following lemma holds.

Figure 4: Simulations accompanying Example 3.1: $\ln$(the number of iterations) vs $\ln(N)$.

**Lemma G.1.** Let $C$ be a group of blocks sampled according to a $\alpha-$proper distribution $P$. Then, for any $y \in \otimes_{r=1}^{R}\mathbb{R}^N$ and $y_{r,i} = 0$, whenever $i \notin S_r$, one has

$$\mathbb{E}_{C\sim P}(\|y_{[C]}\|^2_{2,I(\mu^C)}) = \mathbb{E}_{C\sim P}(\|y_{[C]}\|^2_{2,\theta^P}).$$

*Proof.*

$$\mathbb{E}_{C\sim P}(\|y_{[C]}\|^2_{2,I(\mu^C)}) = \mathbb{E}_{C\sim P}(\sum_{r\in C} \|y_r\|^2_{2,\mu^C}) = \sum_{r\in[R]} \mathbb{E}_{C\sim P}\left[\|y_r\|^2_{2,\mu^C}1_{r\in C}\right]$$

$$= \sum_{r\in[R]} \mathbb{E}\left[1_{r\in C}\mathbb{E}_{C\sim P}\left[\|y_r\|^2_{2,\mu^C}|r \in C\right]\right] = \sum_{r\in[R]} \mathbb{E}\left[1_{r\in C}\|y_r\|^2_{2,\theta_r^P}\right] = \mathbb{E}_{C\sim P}(\|y_{[C]}\|^2_{2,\theta^P}).$$

$\square$

Next, we turn our attention to the proof of the theorem. For this purpose, suppose that $y^* = \arg\min_{y\in\Xi} \|y - y^{(k)}\|_{2,\theta^P}$.

## G.1  Algorithm 1

We start with by establishing the following results.

**Lemma G.2.** It can be shown that

$$\langle \nabla g(y^{(k)}), y^* - y^{(k)}\rangle \overset{1)}{\leq} g(y^*) - g(y^{(k)}) - \frac{1}{N\|\theta^P\|_{1,\infty}}\|y^{(k)} - y^*\|^2_{2,\theta^P}$$

$$\overset{2)}{\leq} \frac{4}{N\|\theta^P\|_{1,\infty} + 2}\left[g(y^*) - g(y^{(k)}) - \frac{1}{2}\|y^{(k)} - y^*\|^2_{2,\theta^P}\right]. \tag{10}$$

*Proof.* From Lemma 3.8 we can infer that

$$\|Ay^{(k)} - Ay^*\|^2_2 \geq \frac{2}{N\|\theta^P\|_{1,\infty}}\|y^{(k)} - y^*\|^2_{2,\theta^P} \Rightarrow$$

$$g(y^*) \geq g(y^{(k)}) + \langle \nabla g(y^{(k)}), y^* - y^{(k)}\rangle + \frac{1}{N\|\theta^P\|_{1,\infty}}\|y^{(k)} - y^*\|^2_{2,\theta^P}, \tag{11}$$

$$g(y^{(k)}) \geq g(y^*) + \langle \nabla g(y^*), y^{(k)} - y^*\rangle + \frac{1}{N\|\theta^P\|_{1,\infty}}\|y^{(k)} - y^*\|^2_{2,\theta^P}. \tag{12}$$

As $\langle \nabla g(y^*), y^{(k)} - y^*\rangle \geq 0$, (12) gives

$$g(y^*) - g(y^{(k)}) \leq -\frac{1}{N\|\theta^P\|_{1,\infty}}\|y^{(k)} - y^*\|^2_{2,\theta^P}. \tag{13}$$

The inequality (11) yields claim 1) in (10). Claim 2) in (10) follows from (13). $\square$

The following lemma is a direct consequence of the optimality of $y_r^{(k+1)}$ for an oblique projection.

**Lemma G.3.**

$$\langle \nabla_r g(y^{(k)}), y_r^{(k+1)} - y_r^* \rangle \leq \langle y_r^{(k)} - y_r^{(k+1)}, y_r^{(k+1)} - y_r^* \rangle_{\theta_r^P}.$$

The following lemma follows from a simple manipulation of the Euclidean norm.

**Lemma G.4.**

$$\frac{1}{2}\|y_r^{(k+1)} - y_r^{(k)}\|_{2,\theta_r^P}^2 = \frac{1}{2}\|y_r^{(k+1)} - y_r^*\|_{2,\theta_r^P}^2 + \frac{1}{2}\|y_r^* - y_r^{(k)}\|_{2,\theta_r^P}^2 + \langle y_r^{(k+1)} - y_r^*, y_r^* - y_r^{(k)} \rangle_{\theta_r^P}$$

$$= -\frac{1}{2}\|y_r^{(k+1)} - y_r^*\|_{2,\theta_r^P}^2 + \frac{1}{2}\|y_r^* - y_r^{(k)}\|_{2,\theta_r^P}^2 + \langle y_r^{(k+1)} - y_r^*, y_r^{(k+1)} - y_r^{(k)} \rangle_{\theta_r^P}$$

Let us analyze next the amount by which the objective function decreases in each iteration. The following expectation is with respect to $C_{i_k} \sim P$.

$$\mathbb{E}\left[g(y^{(k+1)})\right] \tag{14}$$

$$\overset{1)}{\leq} g(y^{(k)}) + \mathbb{E}\left\{ \sum_{r \in C_{i_k}} \left[ \langle \nabla_r g(y^{(k)}), y_r^{(k+1)} - y_r^{(k)} \rangle + \frac{1}{2}\|y_r^{(k+1)} - y_r^{(k)}\|_{2,\mu_r^{C_{i_k}}}^2 \right] \right\}$$

$$\overset{2)}{=} g(y^{(k)}) + \mathbb{E}\left\{ \sum_{r \in C_{i_k}} \left[ \langle \nabla_r g(y^{(k)}), y_r^{(k+1)} - y_r^{(k)} \rangle + \frac{1}{2}\|y_r^{(k+1)} - y_r^{(k)}\|_{2,\theta_r^P}^2 \right] \right\}$$

$$= g(y^{(k)}) + \mathbb{E}\left\{ \sum_{r \in C_{i_k}} \left[ \langle \nabla_r g(y^{(k)}), y_r^* - y_r^{(k)} \rangle + \langle \nabla_r g(y^{(k)}), y_r^{(k+1)} - y_r^* \rangle + \frac{1}{2}\|y_r^{(k+1)} - y_r^{(k)}\|_{2,\theta_r^P}^2 \right] \right\}$$

$$\overset{3)}{\leq} g(y^{(k)}) + \mathbb{E}\left\{ \sum_{r \in C_{i_k}} \left[ \langle \nabla_r g(y^{(k)}), y_r^* - y_r^{(k)} \rangle - \frac{1}{2}\|y_r^{(k+1)} - y_r^*\|_{2,\theta_r^P}^2 + \frac{1}{2}\|y_r^* - y_r^{(k)}\|_{2,\theta_r^P}^2 \right] \right\}$$

$$= g(y^{(k)}) + \alpha \langle \nabla g(y^{(k)}), y^* - y^{(k)} \rangle - \mathbb{E}\left[ \frac{1}{2}\|y_{[C_{i_k}]}^{(k+1)} - y_{[C_{i_k}]}^*\|_{2,\theta^P}^2 \right] + \mathbb{E}\left[ \frac{1}{2}\|y_{[C_{i_k}]}^{(k)} - y_{[C_{i_k}]}^*\|_{2,\theta^P}^2 \right]$$

$$\overset{4)}{=} g(y^{(k)}) + \alpha \langle \nabla g(y^{(k)}), y^* - y^{(k)} \rangle - \mathbb{E}\left[ \frac{1}{2}\|y^{(k+1)} - y^*\|_{2,\theta^P}^2 \right] + \mathbb{E}\left[ \frac{1}{2}\|y^{(k)} - y^*\|_{2,\theta^P}^2 \right]$$

$$\overset{5)}{\leq} g(y^*) - \mathbb{E}\left[ \frac{1}{2}\|y^{(k+1)} - y^*\|_{2,\theta^P}^2 \right] + \left[ 1 - \frac{4\alpha}{N\|\theta^P\|_{1,\infty} + 2} \right] \left\{ g(y^{(k)}) - g(y^*) - \frac{1}{2}\|y^{(k)} - y^*\|_{2,\theta^P}^2 \right\},$$

$$\tag{15}$$

where 1) follows from inequality (8), 2) holds due to Lemma G.1, 3) is a consequence of Lemma G.3 and Lemma G.4, 4) is due to $y_r^{(k+1)} = y_r^{(k)}$ for $r \notin C_{i_k}$, and 5) may be established from (10).

Equation (15) further establishes that

$$\mathbb{E}\left[g(y^{(k+1)}) - g(y^*) + \frac{1}{2}d_{\theta^P}^2(y^{k+1}, \xi)\right] \leq \mathbb{E}\left[g(y^{(k+1)}) - g(y^*) + \frac{1}{2}\|y^{(k+1)} - y^*\|_{2,\theta^P}^2\right]$$

$$\leq \left[1 - \frac{4\alpha}{N\|\theta^P\|_{1,\infty} + 2}\right] \mathbb{E}\left[g(y^{(k)}) - g(y^*) + \frac{1}{2}d_{\theta^P}^2(y^k, \xi)\right].$$

The proof follows by repeating the derivations for all $k$.

## H    Proof of Lemma 3.11

According to the definition of $\theta^P$, we have

$$
\max_{r\in[R]:i\in S_r} \theta^P_{r,i} = \max_{r\in[R]:i\in S_r} \mathbb{E}_{C\sim P}\left[\mu^C_i | r\in C\right]
$$

$$
= \max_{r\in[R]:i\in S_r} \mathbb{E}_{C\sim P}\left[\sum_{r'\in[R]:i\in S_{r'}} 1_{r'\in C} | r\in C\right]
$$

$$
= \max_{r\in[R]:i\in S_r} \sum_{r'\in[R]:i\in S_{r'}} \mathbb{P}_{C\sim P}\left[r'\in C | r\in C\right] \qquad (16)
$$

$$
= \frac{1}{\alpha} \max_{r\in[R]:i\in S_r} \sum_{r'\in[R]:i\in S_{r'}} \mathbb{P}_{C\sim P}\left[r'\in C, r\in C\right]
$$

$$
\geq \frac{1}{\alpha\mu_i} \sum_{r,r'\in[R]:i\in S_r,S_{r'}} \mathbb{P}_{C\sim P}\left[r'\in C, r\in C\right]
$$

$$
= \frac{1}{\alpha\mu_i} \mathbb{E}_{C\sim P}\left[|\{(r,r')\in C\times C : i\in S_r, i\in S_{r'}\}|\right]
$$

$$
= \frac{1}{\alpha\mu_i} \mathbb{E}_{C\sim P}\left[(\mu^C_i)^2\right] \geq \frac{1}{\alpha d_i}\left[\mathbb{E}_{C\sim P}(\mu^C_i)\right]^2 = \frac{1}{\alpha d_i}\left(\sum_C \sum_{r\in[R]:i\in S_r} 1_{r\in C}\mathbb{P}(C)\right)^2
$$

$$
= \frac{1}{\alpha\mu_i}\left(\sum_{r\in[R]:i\in S_r} \mathbb{P}_{C\sim P}[r\in C]\right)^2
$$

$$
= \frac{1}{\alpha\mu_i}(\alpha\mu_i)^2 = \alpha\mu_i.
$$

From (16), we also have $\sum_{r'\in[R]:i\in S_{r'}} \mathbb{P}_{C\sim P}\left[r'\in C | r\in C\right] \geq \mathbb{P}_{C\sim P}\left[r\in C | r\in C\right] = 1$, which proves the claimed result.

## I    Proof of Lemma 3.12

Similar to what was established for (16), one can show that $\theta^P_{r,i} = \sum_{r'\in[R]:i\in S_{r'}} \mathbb{P}_{C\sim P}\left[r'\in C | r\in C\right]$.

Consider next the right hand side of this equation for $\alpha = \frac{K}{R}$. In this case, for some $r$ and some $i\in S_r$, we have

$$
\sum_{r'\in[R]:i\in S_{r'}} \mathbb{P}_{C\sim P}\left[r'\in C | r\in C\right] = \mathbb{P}_{C\sim P}\left[r\in C | r\in C\right] + \sum_{r'\in[R]:i\in S_{r'}, r'\neq r} \mathbb{P}_{C\sim P}\left[r'\in C | r\in C\right]
$$

$$
= 1 + \frac{R}{K} \sum_{r':i\in S_{r'}, r'\neq r} \mathbb{P}_{C\sim P}\left[r'\in C, r\in C\right]
$$

$$
= 1 + \frac{R}{K}(\mu_i - 1)\frac{\binom{R-2}{K-2}}{\binom{R}{K}} = 1 + \frac{K-1}{R-1}(\mu_i - 1).
$$

Therefore, $\theta^P_{r,i} = \frac{K-1}{R-1}\mu_i + \frac{R-K}{R-1}$ when $P$ is a uniform distribution.

## J    Analysis of the Accelerated Coordinate Descend Method

In the ACDM setting, we used the APPROX framework proposed by Fercoq and Richtárik in [30] and adapted it to this particular problem. In the general APPROX framework, the norm in (8) is chosen as follows: consider an arbitrary function $\phi$ with the component-wise smoothness and strong

convexity property. For block $r$, one has $|\nabla_r \phi(x) - \nabla_r \phi(y)| \leq L_r \|x_r - y_r\|_{\nu_r}$, where $\|\cdot\|_{\nu_r}$ is a norm associated with the $r$-th block. If one wants to process multiple blocks simultaneously, say those in a group $C$, one first needs to find a constant $L_C$ such that for any $h$ as defined in (8), it holds that

$$\phi(y + h) \leq \phi(y) + \sum_{r \in C} \langle \nabla_r \phi(y), h_r \rangle + \sum_{r \in C} \frac{L_C}{2} \|h_r\|_{2,\nu_r}^2.$$

The smaller the value of the multiplier $L_C$, the faster the convergence. Typically, $L_C$ lies in $[\max_{r \in C} L_r, \sum_{r \in C} L_r]$.

Recall the smoothness property of $g$ from equation (6). A direct application of APPROX to our problem gives

$$g(y + h) \leq g(y) + \sum_{r \in C} \langle \nabla_r g(y), h_r \rangle + \sum_{r \in C} \frac{\max_{i \in [N]} \mu_i^C}{2} \|h_r\|_2^2.$$

As $(\max_{i \in [N]} \mu_i^C) \geq \mu_j^C$ for all $j \in [N]$, we obtain convergence rates worse than those implied by inequality (8). To actually obtain the guarantees in (8), one needs to dispose with the $\|\cdot\|_2$ norm at the block level and break the blocks into components corresponding to the individual elements. The elements are evaluated independently through the use of the norm $\|\cdot\|_{2,\mu^C}$.

---

**Algorithm 2:  Parallel ACDM for Solving** (4)

**Input**: $\mathcal{B}, \alpha$, some constant $c > 0$
0: Initialize $y^{(0)} \in \mathcal{B}, k \leftarrow 0$
1: $c' \leftarrow \left\lceil (1 + c) \frac{\sqrt{2N\|\theta^P\|_{1,\infty}}}{\alpha} + c \right\rceil$
2: Do the following steps iteratively until the dual gap $< \epsilon$:
3:      If $k = lc'$ for some $l \in \mathbb{Z}$, $z^{(k)} \leftarrow y^{(k)}, \lambda_k \leftarrow 1$
4:      $p^{(k)} \leftarrow (1 - \lambda_k)y^{(k)} + \lambda_k z^{(k)}$
5:      Sample $C_{i_k}$ using some $\alpha$-proper distribution $P$
6:      $z^{(k+1)} \leftarrow z^{(k)}$
7:      For $r \in C_{i_k}$:
8:          $z_r^{(k+1)} \leftarrow \Pi_{\mathcal{B}_r, \theta_r^P}(z_r^{(k)} - \frac{\alpha}{\lambda_k}(\theta_r^P)^{-1} \odot \nabla_r g(p^{(k)}))$
9:      $y^{(k+1)} \leftarrow p^{(k)} + \frac{\lambda_k}{\alpha}(z^{(k+1)} - z^{(k)})$
10:     $\lambda_{k+1} \leftarrow \frac{\sqrt{\lambda_k^4 + 4\lambda_k^2} - \lambda_k^2}{2}$
11:     $k \leftarrow k + 1$
12: Output $y^{(k)}$

---

Similar to the APPROX method [30], the parallel ACDM can also be implemented to avoid full-dimensional vector operations (see Section J.2). The following theorem characterizes the convergence property of Algorithm 2.

**Theorem J.1.** Given $c > 0$, let $c' = \left\lceil (1 + c) \frac{\sqrt{2N\|\theta^P\|_{1,\infty}}}{\alpha} + c \right\rceil$. Consider the iterations $k = lc'$ for $l \in \mathbb{Z}_{\geq 0}$. Then, $y^{(k)}$ of Algorithm 2 satisfies

$$\mathbb{E}\left[ g(y^{(k)} - g(y^*)) \right] \leq \frac{1}{(1 + c)^l} \left[ g(y^{(0)}) - g(y^*) \right].$$

## J.1   Proof of Theorem J.1

We start by establishing a number of background results.

The following lemma is due to the optimality of $z_r^{(k+1)}$.

**Lemma J.2.**

$$\langle \nabla_r g(p^{(k)}), z_r^{(k+1)} - y_r^* \rangle \leq \frac{\lambda_k}{\alpha} \langle z_r^{(k)} - z_r^{(k+1)}, z_r^{(k+1)} - y_r^* \rangle_{\theta_r^P}.$$

Once again, one can easily establish the following result pertaining to the Euclidean norm.

**Lemma J.3.**

$$\frac{1}{2}\|z_r^{(k+1)}-z_r^{(k)}\|_{2,\theta_r^P}^2 = \frac{1}{2}\|z_r^{(k+1)}-y_r^*\|_{2,\theta_r^P}^2 + \frac{1}{2}\|y_r^*-z_r^{(k)}\|_{2,\theta_r^P}^2 + \langle z_r^{(k+1)}-y_r^*, y_r^*-z_r^{(k)}\rangle_{\theta_r^P}$$

$$= -\frac{1}{2}\|z_r^{(k+1)}-y_r^*\|_{2,\theta_r^P}^2 + \frac{1}{2}\|y_r^*-z_r^{(k)}\|_{2,\theta_r^P}^2 + \langle z_r^{(k+1)}-y_r^*, z_r^{(k+1)}-z_r^{(k)}\rangle_{\theta_r^P}.$$

The next result follows from the convexity property of the function $g$.

**Lemma J.4.**

$$\lambda_k\langle\nabla g(p^{(k)}), y^*-z^{(k)}\rangle = \langle\nabla g(p^{(k)}), \lambda_k y^* - \lambda_k z^{(k)}\rangle = \langle\nabla g(p^{(k)}), \lambda_k y^* - (p^{(k)}-(1-\lambda_k)y^{(k)})\rangle$$

$$= \lambda_k\langle\nabla g(p^{(k)}), y^*-p^{(k)}\rangle + (1-\lambda_k)\langle\nabla g(p^{(k)}), y^{(k)}-p^{(k)}\rangle$$

$$\le \lambda_k\left[g(y^*)-g(p^{(k)})\right] + (1-\lambda_k)\left[g(y^{(k)})-g(p^{(k)})\right].$$

We are now ready to analyze the decrease of the objective function in each iteration of Algorithm 2. The expectation in the following equations is performed with respect to $C_{i_k}\sim P$.

$$\mathbb{E}\left[g(y^{(k+1)})\right]$$

$$\overset{1)}{\le} g(p^{(k)}) + \frac{\lambda_k}{\alpha}\mathbb{E}\left\{\sum_{r\in C_{i_k}}\left[\langle\nabla_r g(p^{(k)}), z_r^{(k+1)}-z_r^{(k)}\rangle + \frac{\lambda_k}{2\alpha}\|z_r^{(k+1)}-z_r^{(k)}\|_{2,\mu^C}^2\right]\right\}$$

$$\overset{2)}{=} g(p^{(k)}) + \frac{\lambda_k}{\alpha}\mathbb{E}\left\{\sum_{r\in C_{i_k}}\left[\langle\nabla_r g(p^{(k)}), z_r^{(k+1)}-z_r^{(k)}\rangle + \frac{\lambda_k}{2\alpha}\|z_r^{(k+1)}-z_r^{(k)}\|_{2,\theta_r^P}^2\right]\right\}$$

$$= g(p^{(k)}) + \frac{\lambda_k}{\alpha}\mathbb{E}\left\{\sum_{r\in C_{i_k}}\left[\langle\nabla_r g(p^{(k)}), y_r^*-z_r^{(k)}\rangle + \langle\nabla_r g(p^{(k)}), z_r^{(k+1)}-z_r^*\rangle + \frac{\lambda_k}{2\alpha}\|z_r^{(k+1)}-z_r^{(k)}\|_{2,\theta_r^P}^2\right]\right\}$$

$$\overset{3)}{\le} g(p^{(k)}) + \frac{\lambda_k}{\alpha}\mathbb{E}\left\{\sum_{r\in C_{i_k}}\left[\langle\nabla_r g(p^{(k)}), y_r^*-z_r^{(k)}\rangle - \frac{\lambda_k}{2\alpha}\|z_r^{(k+1)}-y_r^*\|_{2,\theta_r^P}^2 + \frac{\lambda_k}{2\alpha}\|y_r^*-z_r^{(k)}\|_{2,\theta_r^P}^2\right]\right\}$$

$$= g(p^{(k)}) + \lambda_k\langle\nabla g(p^{(k)}), y^*-z^{(k)}\rangle + \frac{\lambda_k^2}{2\alpha^2}\mathbb{E}\left[\|z_{[C_{i_k}]}^{(k)}-y_{[C_{i_k}]}^*\|_{2,\theta^P}^2 - \|z_{[C_{i_k}]}^{(k+1)}-y_{[C_{i_k}]}^*\|_{2,\theta^P}^2\right]$$

$$\overset{4)}{=} g(p^{(k)}) + \lambda_k\langle\nabla g(p^{(k)}), y^*-z^{(k)}\rangle + \frac{\lambda_k^2}{2\alpha^2}\mathbb{E}\left[\|z^{(k)}-y^*\|_{2,\theta^P}^2 - \|z^{(k+1)}-y^*\|_{2,\theta^P}^2\right]$$

$$\overset{5)}{=} g(y^*) + (1-\lambda_k)\left[g(y^{(k)})-g(y^*)\right] + \frac{\lambda_k^2}{2\alpha^2}\left\{\|z^{(k)}-y^*\|_{2,\theta^P}^2 - \mathbb{E}\left[\|z^{(k+1)}-y^*\|_{2,\theta^P}^2\right]\right\},\tag{17}$$

where 1) follows from (8), 2) may be deduced from Lemma G.1, 3) is a consequence of Lemma J.2 and Lemma J.3, 4) is due to the fact that $y_r^{(k+1)}=y_r^{(k)}$ for $r\notin C_{i_k}$, and 5) follows from Lemma J.4.

Based on the definition of $\{\lambda_k\}_{k\ge 0}$, we also have

$$\frac{1-\lambda_k}{\lambda_k^2} = \frac{1}{\lambda_{k-1}^2}, \quad 0 < \lambda_{k+1}\le\lambda_k\le\frac{2}{k+2/\lambda_0} = \frac{2}{k+2}.\tag{18}$$

Hence, combining the above expression with (17), for $k\in[1, \frac{2}{\alpha}\lceil\sqrt{N\|\theta^P\|_{1,\infty}}\rceil+1]$, we have

$$\mathbb{E}\left[\frac{1-\lambda_k}{\lambda_k^2}\left[g(y^{(k)})-g(y^*)\right] + \frac{1}{2\alpha^2}\|z^{(k)}-y^*\|_{2,\theta^P}^2\right]$$

$$= \mathbb{E}\left[\frac{1}{\lambda_{k-1}^2}\left[g(y^{(k)})-g(y^*)\right] + \frac{1}{2\alpha^2}\|z^{(k)}-y^*\|_{2,\theta^P}^2\right]$$

$$\le \mathbb{E}\left[\frac{1-\lambda_{k-1}}{\lambda_{k-1}^2}\left[g(y^{(k-1)})-g(y^*)\right] + \frac{1}{2\alpha^2}\|z^{(k-1)}-y^*\|_{2,\theta^P}^2\right]$$

$$\le\cdots\le\frac{(1-\lambda_0)}{\lambda_0^2}\left[g(y^{(0)})-g(y^*)\right] + \frac{1}{2\alpha^2}\|z^{(0)}-y^*\|_{2,\theta^P}^2.\tag{19}$$

Lemma 3.9 implies the strong convexity property as

$$\|Ay^{(k)} - Ay^*\|_2^2 \geq \frac{2}{N\|\theta^P\|_{1,\infty}} \|y^{(k)} - y^*\|_{2,\theta^P}^2 \Rightarrow$$

$$g(y^{(k)}) - g(y^*) \geq \langle \nabla g(y^*), y^{(k)} - y^* \rangle + \frac{1}{N\|\theta^P\|_{1,\infty}} \|y^{(k)} - y^*\|_{2,\theta^P}^2$$

$$\overset{1)}{\geq} \frac{1}{N\|\theta^P\|_{1,\infty}} \|y^{(k)} - y^*\|_{2,\theta^P}^2. \tag{20}$$

Here, 1) holds since $y^*$ is an optimal solution of $\min_y g(y)$ and thus $\langle \nabla g(y^*), y^{(k)} - y^* \rangle \geq 0$.

Combining (18), (19) and (20), we obtain

$$\mathbb{E}\left[g(y^{(k)}) - g(y^*)\right] \leq \lambda_{k-1}^2 \left[ \frac{1 - \lambda_0}{\lambda_0^2} (g(y^{(0)}) - g(y^*)) + \frac{1}{2\alpha^2} \|y^{(0)} - y^*\|_{2,\theta^P}^2 \right]$$

$$\leq \left(\frac{2}{k+1}\right)^2 \frac{1}{2\alpha^2} \|y^{(0)} - y^*\|_{2,\theta^P}^2$$

$$\leq \left(\frac{2}{k+1}\right)^2 \frac{N\|\theta^P\|_{1,\infty}}{2\alpha^2} (g(y^{(0)}) - g(y^*)).$$

Therefore, for $k = \left\lceil (1+c)\frac{\sqrt{2N\|\theta^P\|_{1,\infty}}}{\alpha} + c \right\rceil$, we have

$$\mathbb{E}\left[ g\left(y^{\left(\left\lceil (1+c)\frac{\sqrt{2N\|\theta^P\|_{1,\infty}}}{\alpha} + c \right\rceil\right)}\right) - g(y^*) \right] \leq \frac{1}{1+c}(g(y^{(0)}) - g(y^*)).$$

For each value of $k = l \times \left\lceil (1+c)\frac{\sqrt{2N\|\theta^P\|_{1,\infty}}}{\alpha} + c \right\rceil$, $l \in \mathbb{Z}_{\geq 0}$, the values $z^{(k)}, \lambda_k$ are reinitialized. Using a similar proof as above, we have

$$\mathbb{E}\left[ g\left(y^{\left((l+1)\times\left\lceil (1+c)\frac{\sqrt{2N\|\theta^P\|_{1,\infty}}}{\alpha} + c \right\rceil\right)}\right) - g(y^*) \right] \leq \frac{1}{1+c}\left[ g\left(y^{\left(l\times\left\lceil (1+c)\frac{\sqrt{2N\|\theta^P\|_{1,\infty}}}{\alpha} + c \right\rceil\right)}\right) - g(y^*) \right].$$

Therefore,

$$\mathbb{E}\left[ g\left(y^{\left(l\left\lceil (1+c)\frac{\sqrt{2N\|\theta^P\|_{1,\infty}}}{\alpha} + c \right\rceil\right)}\right) - g(y^*) \right] \leq \frac{1}{(1+c)^l}(g(y^{(0)}) - g(y^*)).$$

This concludes the proof.

## J.2  Avoiding Full-Dimensional Vector Operations

Algorithm 2 can be implemented without full-dimensional vector operations. In each step, only those coordinates within the blocks in $C$ are updated. Consequently, one only needs to replace $p^{(k)}$ and $y^{(k)}$ with $p^{(k)} = z^{(k)} + \lambda_k^2 u^{(k)}$ and $y^{(k)} = z^{(k)} + \lambda_{k-1}^2 u^{(k)}$, where $u^{(k)}$ is a new variable described in Algorithm 3.

# K  Minimization of $\|\theta^P\|_{1,\infty}$

We first define $\triangle_* \triangleq \max_{r \in [R]} |\{r' \in [R] | S_{r'} \cap S_r \neq \emptyset\}|$, which we use in our subsequent derivations.

As shown in Theorem 3.10 and Theorem J.1, $\|\theta^P\|_{1,\infty}$ plays an important role in the convergence rate of CDMs. Hence, we are interested in identifying the optimal sampling strategy $P$ that minimizes $\|\theta^P\|_{1,\infty}$.

---
**Algorithm 3: Parallel ACDM for Solving Problem (7) (an efficient implementation)**

**Input**: $\mathcal{B}, \alpha$

0: Initialize $z^{(0)} \in \mathcal{B}$, $u^{(0)} \leftarrow 0 \in \mathbb{R}^N$, $k \leftarrow 0$.

1: Do the following steps iteratively until the dual gap $< \epsilon$:

2:   If $k = l \left\lceil (1+c)\sqrt{\frac{2N\|\theta^P\|_{1,\infty}}{\alpha}} + c \right\rceil$ for some $l \in \mathbb{Z}$ and $c > 0$,

$$z^{(k)} \leftarrow z^{(k)} + \lambda_{k-1}^2 u^{(k)}, u^{(k)} \leftarrow 0, \lambda_k \leftarrow 1$$

3:   Sample one set $C_{i_k}$ according to a $\alpha$-proper distribution $P$

4:   For $r \in C_{i_k}$:

5:     $\triangle z_r \leftarrow \arg\min_{\triangle z + z_r^{(k)} \in B_r} \|\triangle z + \frac{\alpha}{\lambda_k}(\theta_r^P)^{-1} \odot \nabla_r g(z^{(k)} + \lambda_k^2 u^{(k)})\|_{2,\theta_r^P}^2$

6:     $z_r^{(k+1)} \leftarrow z_r^{(k)} + \triangle z_r$

7:     $u_r^{(k+1)} \leftarrow u_r^{(k)} + \frac{\lambda_k - \alpha}{\alpha \lambda_k^2} \triangle z_r$

8:   $\lambda_{k+1} \leftarrow \frac{\sqrt{\lambda_k^4 + 4\lambda_k^2} - \lambda_k^2}{2}$

9:   $k \leftarrow k + 1$

10: Output $y^{(k)}$

---

**Algorithm 4: A Greedy Algorithm for Finding a Balanced-Partition Distribution**

**Input**: $\{S_r\}_{r \in [R]}$, $K$

0: Initialize the partition $\mathcal{C} = \{C_i\}_{1 \le i \le m}$, $C_i \leftarrow \emptyset$, vectors $\{\mu^{C_i}\}_{1 \le i \le m}$, $\mu^{C_i} \in \mathbb{R}^N$,
   and $\mu^{\max} \in \mathbb{R}^N$, $\mu^{\max} \leftarrow 0$.

1: For $r$ from 1 to $R$:

2:   For $i$ from 1 to $m$:

3:     If $|C_i| < K$:

4:       $\triangle \mu^{C_i} \leftarrow 0$

5:       For $v$ in $S_r$, if $\mu_v^{C_i}$ is equal to $\mu_v^{\max}$, $\triangle \mu^{C_i} \leftarrow \triangle \mu^{C_i} + 1$

6:     else: $\triangle \mu^{C_i} \leftarrow \infty$

7:   $i^* \leftarrow \arg\min_i \triangle \mu^{C_i}$

8:   $C_{i^*} \leftarrow C_{i^*} \cup \{r\}$

9:   For $v$ in $S_r$, $\mu_v^{C_{i^*}} \leftarrow \mu_v^{C_{i^*}} + 1$, $\mu_v^{\max} \leftarrow \max\{\mu_v^{\max}, \mu_v^{C_{i^*}}\}$.

10: Output $\mathcal{C}$.

---

For this purpose, consider a partition of $[R]$ into $m = \lceil \frac{1}{\alpha} \rceil$ parts $\{C_i\}_{1 \le i \le m}$, such that $|C_i| \in \{K-1, K\}$. We refer to such a partition as a *balanced partition*. In this case, every block $r$ is in exactly one component $C_i$ and $\|\theta^P\|_{1,\infty} = \sum_{v \in [N]} \max_{i \in [m]} \mu_v^{C_i}$. As a result, the problem of minimizing $\|\theta^P\|_{1,\infty}$ is closely related to the so called *equitable coloring problem* first proposed by Meyer [20].

**Definition K.1** (Meyer [20]). Given a graph, an *equitable coloring* is an assignment of colors to the vertices that satisfies the following two properties: no two adjacent vertices share the same color and the number of vertices in any two color classes differs by at most one. Moreover, the minimum number of colors in any equitable coloring is termed the *equitable coloring number*.

Hajnal-Szemerédi's Theorem [25] established one of the most important results in equitable graph coloring: a graph is equitably $k$-colorable if $k$ is strictly greater than the maximum vertex degree. This bound is tight. We can construct a graph based on the incidence structure of DSFM problem so that a vertex corresponds to a component submodular function and two vertices are connected iff the corresponding submodular functions are incident to at least one common point. An equitable coloring of this graph can be used to assign submodular functions of the same color class to a set $C_i$ in $\mathcal{C}$. This guarantees that $\mu_v^{C_i} \le 1$ for all $C_i$ and all $v \in [N]$. Note that the maximal degree of this graph is $\triangle_*$. By directly applying Hajnal-Szemerédi's Theorem, we have the following lemma.

**Lemma K.2.** There exists a balanced-partition distribution $P$ such that $\|\theta^P\|_{1,\infty} = N$, provided that $\lceil \frac{1}{\alpha} \rceil \ge \triangle_* + 1$.

As in many applications, such as image segmentation [9], the value of $\triangle_*$ is small, and hence using a balanced-partition instead of one obtained through sampling uniformly at random may produce significantly better results. Unfortunately, finding the equitable coloring number is an NP-hard

problem; still, a polynomial time algorithm for finding $\triangle_* + 1$ equitable colorings was described in [31], with complexity $O(\triangle_* R^2)$. We describe a greedy algorithm that outputs a balanced-partition distribution and aims to minimize $\|\theta^P\|_{1,\infty}$ in Algorithm 4. According to our experimental results, the sampling strategy $P$ found by Algorithm 4 works better than sampling uniformly at random.

## L   Using Weighted Proximal Terms

The AP and RCDM solvers discussed in the main text are designed to solve the convex optimization (2), but also produce a solution to the discrete optimization problem (1). To solve the discrete optimization problem (1), another convex optimization formulation may be considered instead:

$$\min_{x \in \mathbb{R}^N} \sum_{r \in [R]} f_r(x) + \frac{1}{2} \|x\|_{2,w}^2, \tag{21}$$

where the choice of $w \in \mathbb{R}^N_{>0}$ will be described later. By using the arguments in [32] or in Chapter 8.1-8.2 of [12], we know that the solution of the discrete optimization problem (1) can be obtained as $S = \{i \in [N] | x_i^* > 0\}$, where $x^*$ is a solution of (21).

Next, we describe how a proper choice of $w$ allows one to avoid compute oblique projections in the AP and parallel CDM algorithms. If oblique projections are allowed, a good choice for $w$ may also decrease the computational complexities listed in Table 1. The results obtained based on weighted proximal terms are summarized in Table 2.

| | Using Orthogonal Projection $\Pi_{\mathcal{B}_r}(\cdot)$ | |
|---|---|---|
| | The Value of $w$ | Complexity |
| AP | $w = \mu$ | $O(N\|\mu\|_1 \frac{R}{K})$ |
| RCDM | $w = \frac{R-K}{R-1}1 + \frac{K-1}{R-1}\mu$ | $O\left( \left( \frac{R-K}{R-1}N^2 + \frac{K-1}{R-1}N\|\mu\|_1 \right) \frac{R}{K} \right)$ |
| ACDM | $w = \frac{R-K}{R-1}1 + \frac{K-1}{R-1}\mu$ | $O\left( \left( \frac{R-K}{R-1}N^2 + \frac{K-1}{R-1}N\|\mu\|_1 \right)^{\frac{1}{2}} \frac{R}{K} \right)$ |
| | Using Oblique Projection $\Pi_{\mathcal{B}_r, w^{1/2}}(\cdot)$ | |
| | The Value of $w$ | Complexity |
| AP | $w = \mu^{\frac{1}{2}}$ | $O(\|\mu^{\frac{1}{2}}\|_1^2 \frac{R}{K})$ |
| RCDM | $w = \left( \frac{R-K}{R-1}1 + \frac{K-1}{R-1}\mu \right)^{\frac{1}{2}}$ | $O\left( \left\| \left( \frac{R-K}{R-1}1 + \frac{K-1}{R-1}\mu \right)^{\frac{1}{2}} \right\|_1^2 \frac{R}{K} \right)$ |
| ACDM | $w = \left( \frac{R-K}{R-1}1 + \frac{K-1}{R-1}\mu \right)^{\frac{1}{2}}$ | $O\left( \left\| \left( \frac{R-K}{R-1}1 + \frac{K-1}{R-1}\mu \right)^{\frac{1}{2}} \right\|_1 \frac{R}{K} \right)$ |

Table 2: New complexity results based on weighted proximal terms: here, complexity refers to the required number of iterations needed to achieve an $\epsilon-$optimal solution (the dependence on $\epsilon$ is the same for all algorithms and hence omitted). As before, $K$ is the parallelization parameter and it equals the number of min-norm points problems that are solved within each iteration; $K = 1$ reduces to the sequential case.

We now analyze the new objective (21) in more detail. The proof techniques used in the main text carry over to the setting involving weighted proximal terms.

By using a dual strategy similar to those described in Lemma 2.1 and Lemma 3.2, we arrive at the dual formulation of problem (21) described in the next lemma. Note that the derivation of (L.1) takes into account the underlying incidence relations.

**Lemma L.1.** The dual problem of (21) reads as

$$\min_{a,y} \|a - y\|_{2,I(w^{-1}\odot\mu)}^2 \quad \text{s.t.} \quad y \in \mathcal{B}, Aa = 0, \text{ and } a_{r,i} = 0, \ \forall(r,i) : i \notin S_r, r \in [R]. \tag{22}$$

Moreover, problem may be written in a more compact form as

$$\min_y \|Ay\|_{2,w^{-1}}^2 \quad \text{s.t.} \quad y \in \mathcal{B}. \tag{23}$$

For both problems, the primal and dual variables are related according to $x = -w^{-1} \odot Ay$.

## L.1 The Incidence Relations AP (IAP) Method for Solving (L.1)

The steps of the IAP method are listed in Algorithm 5.

---
**Algorithm 5: The IAP Method for Solving** (L.1)
---
0: For all $r$, initialize $y_r^{(0)} \in \mathcal{B}_r$, and $k \leftarrow 0$
1: In iteration $k$:
2:    For all $r \in [R]$:
3:       $a_{r,i}^{(k+1)} \leftarrow y_{r,i}^{(k)} - \mu_i^{-1}(Ay^{(k)})_i$ for all $i \in S_r$
4:       $y_r^{(k+1)} \leftarrow \Pi_{\mathcal{B}_r, w^{-1}\odot\mu}(a_r^{(k+1)})$

---

The convergence properties of Algorithm 5 can be characterized similarly as those of IAP for solving (5). The latter relies on a finite upper bound for $\kappa_* \triangleq \sup_{y \in \mathcal{Z} \cup \mathcal{B}/\Xi} \frac{d_{I(w^{-1}\odot\mu)}(y,\Xi)}{\max\{d_{I(w^{-1}\odot\mu)}(y,\mathcal{Z}), d_{I(w^{-1}\odot\mu)}(y,\mathcal{B})\}}$.

**Lemma L.2.** One has $\kappa_* \leq \sqrt{\frac{\|w^{-1}\odot\mu\|_1 \|w\|_1}{2}} + 1$. When $w = \mu$, $\kappa_* \leq \sqrt{\frac{N\|\mu\|_1}{2}} + 1$.

*Proof.* The result follows using the same strategy as the one used to prove Lemma 3.4. Note that when using Lemma 3.1, one should set $\theta$ to $I(w^{-1} \odot \mu)$ and replace $w$ by $w^{-1}$. $\qquad\square$

By setting $w = \mu$, Step 4 of Algorithm 5 reduces to orthogonal projections. In this case, based on Lemma L.2, Algorithm 5 requires $O(N\|\mu\|_1 \log\frac{1}{\epsilon})$ iterations to achieve an $\epsilon$-optimal solution. By setting $w = \mu^{\frac{1}{2}}$ for all $i \in [R]$, Step 4 of Algorithm 5 reduces to the projections $\Pi_{\mathcal{B}_r, w^{\frac{1}{2}}}(\cdot)$. In this case, Algorithm 5 requires $O\left(\left\|\mu^{\frac{1}{2}}\right\|^2 \log\frac{1}{\epsilon}\right)$ iterations to achieve an $\epsilon$-optimal solution. The latter result is slightly better because $\left\|\mu^{\frac{1}{2}}\right\|^2 \leq N\|\mu\|_1$.

## L.2 A Parallel RCD Method for Solving (23) with Uniform Sampling Strategies

As discussed in Section 3.3, RCDM strongly depends on an $\alpha$-proper distribution $P$ that characterizes the parallel coordinate sampling strategy. In what follows, we choose $P$ to be a uniform distribution. From Lemma 3.12, we know that when $P$ is uniform, one has $\theta_r^P = \frac{K-1}{R-1}\mu + \frac{R-K}{R-1}\mathbb{1}$ for all $r \in [R]$, where $K$ denotes the number of projections computed in parallel as part of each iteration. In Algorithm 1, $\theta_r^P$ defines the normed space over which to minimize $g(y)$. As our goal is to minimize $g_w(y) = \frac{1}{2}\|Ay\|_{2,w^{-1}}^2$, the vector used to define the normed space is

$$\nu = w^{-1} \odot \theta_r^P = w^{-1} \odot \left(\frac{K-1}{R-1}\mu + \frac{R-K}{R-1}\mathbb{1}\right).$$

The parallel RCDM procedure in this setting is described in Algorithm 6.

---
**Algorithm 6: Parallel RCDM with Uniform Sampling for Solving** (23)
---
**Input**: $\mathcal{B}, K$
0: Initialize $y^{(0)} \in \mathcal{B}, k \leftarrow 0$
1: Do the following steps iteratively until the dual gap $< \epsilon$:
2:    Uniformly sample $C_{i_k} \subseteq [R]$ so that $|C_{i_k}| = K$.
3:    For $r \in C_{i_k}$:
4:       $y_r^{(k+1)} \leftarrow \Pi_{\mathcal{B}_r, \nu}(y_r^{(k)} - (\nu^{-1}) \odot \nabla_r g_w(y))$
5:    Set $y_r^{(k+1)} \leftarrow y_r^{(k)}$ for $r \notin C_{i_k}$
6:    $k \leftarrow k + 1$
7: Output $y^{(k)}$

---

Similarly to what was done in Lemma 3.9, we can establish weak strong convexity of $g_w(y)$ with respect to the norm $\|\cdot\|_{2,\nu}$ by invoking Lemma 3.1.

**Lemma L.3.** For any $y \in \mathcal{B}$, let $y^* = \arg\min_{\xi \in \Xi} \|\xi - y\|_{2,\nu}^2$. Then,

$$\|Ay - Ay^*\|_{2,w^{-1}}^2 \geq \frac{2}{\|w\|_1 \|\nu\|_1} \|y - y^*\|_{2,\nu}^2.$$

Therefore, using a strategy similar to the one outlined in the proof of Theorem 3.10, the convergence rates of Algorithm 6 can be derived as summarized in the next theorem.

**Theorem L.4.** At each iteration of Algorithm 6, $y^{(k)}$ satisfies

$$\mathbb{E}\left[g_w(y^{(k)}) - g_w(y^*) + \frac{1}{2}d_{I(\nu)}^2(y^k, \xi)\right]$$

$$\leq \left[1 - \frac{4K}{R(\|w\|_1 \|\nu\|_1 + 2)}\right]^k \left[g_w(y^{(0)}) - g_w(y^*) + \frac{1}{2}d_{I(\nu)}^2(y^0, \xi)\right].$$

By setting $w = \frac{K-1}{R-1}\mu + \frac{R-K}{R-1}\mathbf{1}$, we reduce the projections in Step 4 of Algorithm 6 to orthogonal projections. In this case, based on Theorem L.4, Algorithm 6 requires $O\left(\left(\frac{K-1}{R-1}N\|\mu\|_1 + \frac{R-K}{R-1}N^2\right)\frac{R}{K}\log\frac{1}{\epsilon}\right)$ iterations to achieve an $\epsilon$-optimal solution.

By setting $w = \left(\frac{K-1}{R-1}\mu + \frac{R-K}{R-1}\right)^{1/2}$ for all $i \in [R]$, the projections in Step 4 of Algorithm 6 reduce to oblique projections $\Pi_{\mathcal{B}_r, w^{\frac{1}{2}}}(\cdot)$. In this case, Algorithm 6 requires $O\left(\left\|\left(\frac{K-1}{R-1}\mu_i + \frac{R-K}{R-1}\right)^{1/2}\right\|_1^2 \log\frac{1}{\epsilon}\right)$ iterations to achieve an $\epsilon$-optimal solution, which is slightly better than the previous case. The accelerated methods can be analyzed in the same manner.

## L.3 Simulations

We now describe simulation results that empirically evaluate Algorithms 5 and 6. The DSFM problem is designed as follows. We consider $N = 100$ vertices. The unary potentials of different elements are iid standard Gaussian variables. We construct a network over these vertices based on the Barabási-Albert model (BA) [33], initialized with a single edge between vertices 1 and 2. Each edge in the network gives a pairwise potential for the corresponding vertices. We use the BA model so that the number of incidence relations corresponding to different vertices vary to a large extent. As we are using weighted proximal terms, the continuous objectives are not consistent for different $w$. However, here, we are only interested in generating solutions for the discrete problem (1) and thus regard the discrete gap $\nu_d$ as the relevant metric for characterizing convergence properties. The following results are obtained from 100 independent experiments.

In IAP (Algorithm 5), we set $w \in \{1, \mu, \mu^{1/2}\}$, corresponding to three cases: *unweighted proximal term + oblique projections*, *weighted proximal term + orthogonal projections*, *weighted proximal term + oblique projections*, respectively. In RCDM-U (Algorithm 6), we set $w \in \{1, \frac{K-1}{R-1}\mu + \frac{R-K}{R-1}\mathbf{1}, (\frac{K-1}{R-1}\mu + \frac{R-K}{R-1}\mathbf{1})^{1/2}\}$, corresponding to the same three cases. We control the number of parallel projection operations in each iteration by choosing $K \in \{10, 20, 30, 40, 50\}$. Figure 5 shows the convergence curve of the discrete gap for different solvers and different choices of $w$. We only plotted results for $K = 10, 50$ as other values of $K$ produce similar patterns. For both IAP and RCDM-U, when $w$ corresponds to the weighted proximal term + orthogonal projections case, we obtain the best convergence rates. The value $w = 1$, corresponding to the case unweighted proximal term + oblique projections, results in the worst convergence rates. Albeit somewhat inconsistent with the results listed in Table 2, the simulations simply imply that using weighted proximal terms can reduce the complexity of the algorithms at hand and that the weighted proximal term with orthogonal projections in the inner loop may represent the best choice in practice.

In Table 3, we also list the number of iterations needed by different solvers to obtain a solution for the discrete problem (1). Again, the $w$ corresponding to the weighted proximal term + orthogonal projections case results in the smallest number of iterations, while the $w$ corresponding to the unweighted proximal term + oblique projections case results in the largest number of iterations. Note that as $K$ increases, the number of iterations $\times K/R$ in IAP does not change as IAP is fully parallelizable, while the number of operations in RCDM-U slightly increases due to the overlapping incidence sets of different submodular functions.

Figure 5: Simulations for Algorithm 5 and 6: $\log_{10}(\text{discrete gap})$ vs (number of iterations $\times K/R$).

| Solvers | $w$ | $K=10$ | | $K=20$ | | $K=30$ | | $K=40$ | | $K=50$ | |
|---|---|---|---|---|---|---|---|---|---|---|---|
| | | MN | MD | MN | MD | MN | MD | MN | MD | MN | MD |
| IAP | 1 | 109 | 103 | 109 | 103 | 109 | 103 | 109 | 103 | 109 | 103 |
| | $\mu$ | 43 | 34 | 43 | 34 | 43 | 34 | 43 | 34 | 43 | 34 |
| | $\mu^{1/2}$ | 59 | 50 | 59 | 50 | 59 | 50 | 59 | 50 | 59 | 50 |
| RCDM-U | 1 | 27 | 22 | 34 | 28 | 43 | 38 | 51 | 46 | 54 | 49 |
| | $\frac{K-1}{R-1}\mu + \frac{R-K}{R-1}1$ | 22 | 17 | 25 | 20 | 29 | 24 | 32 | 24 | 33 | 25 |
| | $\left(\frac{K-1}{R-1}\mu + \frac{R-K}{R-1}1\right)^{1/2}$ | 25 | 19 | 28 | 23 | 33 | 28 | 37 | 31 | 38 | 32 |

Table 3: The number of iterations $\times K/R$ needed to find an optimal solution to the discrete problem (1). MN: mean; MD: median.

## M  Supplementary experiments

**Semi-supervised learning over hypergraphs**. We also evaluate the proposed approaches over the 20Newsgroups from the University of California Irvine (UCI) data repository. This dataset is used as a benchmark example for evaluating semisupervised learning algorithms over hypergraphs [34, 35]. Here, for simplicity, we focused on binary classification tasks and thus paired the four 20Newsgroups classes, so that one group includes "Comp." and "Sci", and the other one includes "Rec." and "Talk". The 20Newsgroups dataset consists of categorical features and we adopt the same approach as the one described in [34] to construct hyperedges: each feature corresponds to one hyperedge and contributes one submodular function to the decomposition. Hence, 20Newsgroups contains $N = 16242$ elements and $R = 100$ submodular functions.

In the experiments for 20Newsgroups, we uniformly at random picked 200 elements and set their corresponding components in $x_0$ of equation (9) to the true labels and set all other entries to zero. Figure 6 shows the results of the experiments pertaining to 20Newsgroup. We compared the convergence rate of different algorithms for different values of the parameter $\alpha \in \{0.02, 0.1\}$. The value on the horizontal axis, # iterations $\times \alpha$, equals the total number of projections, scaled by $R$. The results are averaged over 10 independent experiments. Once again, we observe that CD-based methods outperform AP-based methods. ACDM-U offers the best performance among all CD-based methods and IAP significantly outperforms AP. Similarly, RCDM-G has better performance than RCDM-U, due to the use of the greedy algorithm for the sampling procedure.

Figure 6: 20Newsgrounp: Smooth/discrete gap vs the (number of iterations $\times\alpha$).