[Reviews · NeurIPS 2018]

Reviewer 1



The authors consider the problem of submodular minimization under factorization and incidence assumptions. Specifically, they assume that the function decomposes into terms that depend only on a subset of the variables. They consider the minimum norm problem and two algorithms - alternating minimization (AP) and coordinate descent (CD). In both cases, they improve the old bounds in a pretty natural way - they replace NR by the sum of the neighbourhoods of the components. AP under incidence constraints has been also studied by Scalable Variational Inference in Log-supermodular Models , Josip Djolonga, Andreas Krause, ICML 2015, who also extend the results from Nishihara et al, and arrive at exactly the same problem as in this paper (they prove only under the assumption of a regular graph though). The fact that one scales by the inverse of the number of components (i.e. the skewed norms) is not surprising, as similar problems also appear in classical dual decomposition. Their randomized CD algorithm is novel, as they showed that one can not easily extend the results from Ene et al. I find that the intuition behind their approach is not well explained and the discussion could certainly be improved. I am not sure about the applicability of these models though. For example, in the experimental section they split into 1065 components, but this is necessarily as only two can suffice -- all vertical and all horizontal edges. Similarly, the superpixels typically do not overlap -- if one does several layers they will overlap, but this is typically on the order of 2 to 3. I think if they implement it that way, there won't be any benefits when compared to the classical AP and RCD methods. Moreover, they only implement AP, while Jegelka et al. show that a reflection based approaches obtains much faster convergence. Finally, they measure certified duality gap vs # of iterations - I think the x-axis should be wall-clock time, as the comparison is unfair because different methods do varying number of projections per iteration. Questions --- * In the experiments for the superpixel potentials you do not have to use divide and conquer, as a single sort + isotonic regression would suffice, as explained in Lim, Cong Han, and Stephen J. Wright. "Efficient Bregman projections onto the permutahedron and related polytopes." AISTATS 2016. * l73, 74 - Maybe note that { i | x_i > 0 } and { i | x_i >= 0 } are both in the lattice of minimizers. * l121. Can't you say that I(w) consists of R copies of w? It can be a bit confusing in the way it is stated, especially with the variable r appearing twice. * l234. Not clear why convergence is guaranteed. Aren't you simply showing that you minimize an upper bound? More discussion could certainly help in this section. * Def 3.8 Maybe say "We say that P is an alpha-proper for some \alpha\in(0,1) if ....". A better name might be alpha-sparse? * Example 3.1. - Perhaps note that this is exactly a cut function on a chain graph? * Spell out maybe in more detail why the inequality l* < ... holds? What is the y^* from Def 3.6 for this y? * l230: unclear if you mean h_{r,i} is an indicator vector on coordinate {r, i} or if for all r\in C only y_{r,i} is nonzero, or i h_{r,*} for r\in C are the only non-zero blocks. * You can cheaply improve the certificates using isotonic regression, as explained in Bach, [p. 281 ,"Primal candidates from dual candidates."]. Post rebuttal --- In Djolonga et al., they analyze the exactly same problem, even though the motivation is through L-Field. They specifically show that the problems are equivalent and use the same weighted projections, despite proving only the regular case. The authors did not respond to how the results would look like if the natural decomposition is done for the images. I expect their improvements to significantly diminish for this setting. I think the paper is interesting and the setting they consider is natural, although I do not think the practical benefits are well supported by the experiments, as the instances chosen have very natural and simple decompositions.

Reviewer 2



The paper presents new and improves existing algorithms for decomposable submodular function minimization (DSFM) by exploiting certain structures known as incidence relations between the ground set and the component submodular functions. After making this key observation about subsets of variables that are relevant and directly affect the value of any given component function, the paper theoretically and experimentally shows that known algorithms like alternating projections in [14] can be improved to achieve faster convergence rates. The authors further show that though the complexity of sequential coordinate descent (CD) methods cannot be improved using incidence relations, these methods can be parallelized and the convergence rates of parallel algorithms depends on these incidence relations. The paper is very well written and I advocate for acceptance of this submission to NIPS. Typo: 1. The conditions in line 108 of page 3 should be to verify whether an element i is NOT incident to F. Consider a trivial example where F(S) = 0 for every set S. The conditions in line 108 are satisfied but none of the elements i are incident as per the condition in line 106. Minor concerns: 1. The authors should discuss the computational complexity for determining the set S_r as it directly determines the quantity ||\mu||_1 critical to their analysis. 2. The conditions stated in line 108 needs more explanation.

Reviewer 3



Summary: The paper considers the decomposable submodular function minimization (DSFM) problem. They define incidence relation between the submodular functions and the elements of the ground set to propose an alternating projection algorithm and random co-ordinate descent algorithm that can be used to solve the DSFM problem. They show their results on images to demonstrate that they need lesser number of projections than the state of art. Pros: - The bounds on the number of iterations required to solve the problem for an epsilon optimal solution are promising. - They define incidence relation between submodular functions and the elements of the ground set that they use to propose these fast algorithms. However, this uses the basic idea that alternating projection onto orthogonal low dimensional polytopes leads to quicker convergence as shown by Nishihara et al, Incidence relationship encodes the orthogonality of the base polytopes instrinsically. This is my understanding. - They show the experiments on oct and smallplant images and compare them with the state of art algorithms. Cons/Clarifications: - The authors did comment about the standard MNP that is used to project onto the base polytope and the projection algorithm they use. I would like them to elaborate further if it is possible to get exact projections. I believe exact projections may only be achieved using divide and conquer that assumes and Submodular Functions Minimization oracle for each F_r. - I remember to have used the code by Prof. Stefanie Jegelka to solve the discrete and continuous problems using Alternating Projection and Douglas Rachford. They seems to have converged much quicker on the same datasets as compared to the results shown. It would be helpful for me to understand if the code was modified to have a different alternating projection. If so would it be possible to compare the original code with the proposed ones on the same dataset. Update after Author Response: I apologize I agree that MNP is exact but expensive. However, the results section does not seem to do a fair comparison of different algorithms. Therefore, I would like to stick to my earlier rating as I believe, the theoretical contribution is not completely novel and the results section is not strong enough.

Reviewer 4



I am not an expert on submodular optimization, I therefore only focussed on the convex optimization problem and the coordinate descent algorithm part. Algorithm 1 is a way how to implement the standard coordinate descent algorithm when probabilities of choosing coordinates are coming from given distribution P. the description of the algorithm is following the standard in the literature